# Analysis of the chemical constituents and their metabolites in *Orthosiphon stamineus* Benth. via UHPLC-Q exactive orbitrap-HRMS and AFADESI-MSI techniques

**Jianting Ouyang**[1,2,3,4], **Danyao Lin**[1,2,3,4], **Xuesheng Chen**[1,2,3,4], **Yimeng Li**[1,2,3,4], **Qin Liu**[1], **Delun Li**[1,2,3,4], **Haohao Quan**[1,2,3,4], **Xinwen Fu**[1,2,3,4], **Qiaoru Wu**[1,2,3,4], **Xiaowan Wang**[3], **Shouhai Wu**[2,3], **Chuang Li**[1,2,3,4]*, **Yi Feng**[2,4,5]*, **Wei Mao**[1,2,3,4]*

**1** The Second Clinical College, Guangzhou University of Chinese Medicine, Guangzhou, China, **2** State Key Laboratory of Dampness Syndrome of Chinese Medicine, The Second Affiliated Hospital of Guangzhou University of Chinese Medicine, Guangzhou, China, **3** Department of Nephrology, The Second Affiliated Hospital of Guangzhou University of Chinese Medicine (Guangdong Provincial Hospital of Chinese Medicine), Guangzhou, China, **4** Guangdong Provincial Academy of Chinese Medical Sciences, Guangzhou, China, **5** Department of Pharmacokinetics of Guangzhou University of Chinese Medicine (Guangdong Provincial Hospital of Chinese Medicine), Guangzhou, China

* lichuang@gzucm.edu.cn (CL); fy1633@126.com (YF); maowei@gzucm.edu.cn (WM)

**Data Availability Statement:** All relevant data are within the manuscript and its Supporting Information files.

## Abstract

### Background

Known for its strong diuretic properties, the perennial herbaceous plant *Orthosiphon stamineus* Benth. is believed to preserve the kidney disease. This study compared the boiling water extract with powdered *Orthosiphon stamineus* Benth. and used a highly sensitive and high resolution UHPLC-Q-Exactive-Orbitrap-HRMS technology to evaluate its chemical composition.

### Results

Furthermore, by monitoring the absorption of prototype components in rat plasma following oral treatment, the beneficial ingredients of the *Orthosiphon stamineus* Benth. decoction was discovered. Approximately 92 substances underwent a preliminary identification utilizing relevant databases, relevant literature, and reference standards. As the compound differences between the powdered *Orthosiphon stamineus* Benth. and its water decoction were analyzed, it was found that boiling produced additional compounds, 48 of which were new. 45 blood absorption prototype components and 49 OS metabolites were discovered from rat serum, and a kidney tissue homogenate revealed an additional 28 prototype components. Early differences in the distribution of ferulic acid, cis 4 coumaric acid, and rosmarinic acid were shown using spatial metabolomics. It was elucidated that the renal cortex region is where rosmarinic acid largely acts, offering a theoretical foundation for further studies on the application of OS in the prevention and treatment of illness as well as the preservation of kidney function.

**Funding:** This work was supported by National Natural Science Foundation of China 82074376 (Wei Mao) and 82374388(Chuang Li), the Specific Fund of State Key Laboratory of Dampness Syndrome of Chinese Medicine SZ2021ZZ50(Wei Mao), SZ2021ZZ100(Chaung Li). Guangdong Provincial Science and Technology Project 2022A1515012051(Wei Mao). Proof Wei Mao and Chuang Li main involvement in this manuscript are study design and decision to publish.

## Significance

In this study, UHPLC–Q Exactive Orbitrap–HRMS was employed to discern OS's chemical composition, and a rapid, sensitive, and broad-coverage AFADESI-MSI method was developed to visualize the spatial distribution of compounds in tissues.

## 1. Introduction

*Orthosiphon stamineus* Benth. (Lamiaceae, which has been checked with http://www.worldfloraonline.org on 7 December 2023) (OS) which belongs to the Labiatae family, distinguished as one of the most expansive and distinctive angiosperm families globally is an established perennial herb [1–3], with a vast distribution in tropical and subtropical regions [4]. This includes Southeast Asian countries such as Indonesia, Malaysia, Thailand, Vietnam, Myanmar, and the Philippines [5, 6], southern China [7], India [8], Australia [9], among others. Apart from *Orthosiphon stamineus* Benth., it is also scientifically known as Clerodendranthus spicatus (Thunb) c. y. wu and Orthosiphon aristatus (Blume) Miq. [10, 11]. Chinese "Shencha" is another common name for this plant, specifically referring to the ground-up portion of *Orthosiphon stamineus* Benth. It's also known as "Cat's whiskers" [12], "Misai Kucing" [13], "Java tea" [14], and "kumis kucing" [15] in several Southeast Asian nations. O. stamineus is renowned for its powerful diuretic effect, surpassing most natural diuretics in efficacy. Existing literature highlights its significant contribution to hyperuricemia nephropathy treatment, and its proven renal protective effects [16]. However, specific research identifying the compound or compounds responsible remains absent.

The analysis of chemical components in traditional Chinese medicine, particularly the qualitative and structural identification of active ingredients, can elucidate the fundamentals of the drug's effectiveness and offer a scientific foundation for comprehending the mechanism of illness prevention and treatment. Ultra-performance liquid chromatography electrospray ionization mass spectrometry (UPLC-ESI-MS/MS) has emerged as a potent analytical tool for detecting natural product components, owing to its high sensitivity, low solvent consumption, and rapid speed [17, 18]. However, it is acknowledged that only the components absorbed into the blood circulation can function as active ingredients and exert a therapeutic effect [19, 20]. Consequently, it is vital to study the *in vivo* absorption prototypes and metabolites of traditional Chinese medicine alongside investigating its chemical components. Considering the unclear safety profiles of most herbs in the human body, the fact that numerous herbs yield different results *in vitro* and *in vivo*, attention must be directed towards understanding the metabolism of herbs and considering both systems to effectively prevent adverse drug events [21].

Mass spectrometry imaging (MSI) is an impactful label-free technique that offers detailed maps of numerous molecules in complex samples with high sensitivity and subcellular spatial resolution [22]. Secondary ion mass spectrometry (SIMS) and matrix-assisted laser desorption ionization (MALDI) are two significant MSI methods usually performed in a vacuum [23]. Particularly, MALDI-MSI yields ultra-high spatial resolution as low as 600nm and exhibits information sensitivity [24, 25]. Ambient ionization mass spectrometry techniques such as desorption electrospray ionization (DESI) and laser ablation electrospray ionization (LAESI) have been developed for direct tissue imaging [26–29]. Air flow-assisted desorption electrospray ionization mass spectrometry imaging (AFADESI-MSI) is a technique grounded in DESI [30]. Besides harnessing the benefits of DESI-MSI, AFADESI-MSI also accomplishes

extensive coverage of examined metabolites, enabling the detection of thousands of molecules concurrently in non-targeted experiments [31]. Moreover, it permits whole body section imaging [32]. Presently, ambient ionization MSI has been extensively employed in cancer diagnosis and can be applied to any disease pathology necessitating tissue analysis, such as the histopathology of kidney diseases, infectious diseases, transplants, skin diseases, fertility, and metabolic diseases [33]. However, given the complexity and heterogeneity of tissue samples, developing a method with broad metabolite coverage, high sensitivity, a wide dynamic range, and high specificity remains challenging. In this study, a rapid, sensitive, and broad-coverage AFADESI-MSI method was developed to visualize the spatial distribution of compounds in tissues.

In this study, UHPLC–Q Exactive Orbitrap–HRMS was employed to discern OS's chemical composition. Sixty reference standards were used to verify the compounds contained in OS, and to analyze the differences in compounds extracted between OS water decoction and OS powder methanol extraction. Subsequent to this fundamental analysis, the compounds of OS decoction in serum and kidney tissue homogenate were identified. Based on the fingerprint of OS, the metabolism of OS in the serum of SD rats was extrapolated. Moreover, the AFADESI-MSI technique was applied to uncover the distribution of compounds such as rosmarinic acid in the kidneys of SD rats. This method paves the way for a deeper investigation into the OS target and *in vivo* mechanism.

## 2. Materials and methods

### 2.1 Reagents and materials

OS was purchased from Zisun Medicine (guangzhou, China, lot:200901), the country of origin in Guangxi, with implementation standard: Fujian Province Code for the processing of traditional Chinese Medicine, 1988 edition. The dry above ground portion of the OS is used. Purchase details for reference standard can be found in the S1 Table. The purity of all reference standards is greater than 98%. HPLC-MS grade acetonitrile Thermo Fisher Scientific (US) and HPLC grade methanol were supplied by Merck (Darmstadt, Germany). Formic acid was supplied from Aladdin Chemistry (Shanghai, China). Ultrapure deionized water was supplied from Watsons (Hong Kong, China).

### 2.2 Preparation of lyophilized powder of OS decoction and methanol extract of OS powder

The process to prepare lyophilized powder of OS water decoction involved the following steps: Firstly, 200g of OS were weighed and placed in a glass bottle with a 5L electric heating cover, along with 2400ml of pure water. After soaking for 30 minutes, the mixture was heated to the boiling point, after which the flame was reduced and the mixture was simmered for 2 hours. A condensing tube was put in place to minimize liquid evaporation loss. Subsequent to this, the mixture was filtered, and 2000ml of pure water was added for a second heating cycle. This involved bringing it to boil and then reducing the flame to allow it to simmer for 3 hours. The mixture was then filtered again, and the resulting decoctions from the two heating cycles were combined and filtered. A rotary evaporator was used to evaporate and concentrate the mixture to 160ml, yielding 1.25g/ml of OS decoction. The aforementioned water decoction was placed in a disposable bowl and freeze-dried in a vacuum freeze-drying machine. The final concentration of the freeze-dried OS powder was determined by comparing weights before and after freeze-drying to be 1g/ml. For subsequent use, the corresponding weight of lyophilized powder was weighed directly and added to a 1:1 (v:v) methanol water mixture. This was mixed and

sonicated in an ultrasonic machine for 30 minutes, followed by sample injection through a pinhole filter membrane for detection.

The process for preparing a methanol extract of OS begins with the creation of OS powder, which is achieved by grinding OS in a grinder and subsequently sieving it. Hereafter, 10mg of the OS powder was added directly to a 1:1 *(v/v)* methanol water mixture. Following a thorough mix, the solution is placed in an ultrasonic machine for 30 minutes. Upon completion of the ultrasound, the sample is filtered through a pinhole filter membrane for injection.

## 2.3 Animals

Male SD rats (200 ± 20 g) were obtained from the Experimental Animal Center of Guangzhou University of Chinese Medicine (Guangzhou, China). All animal experiments were approved by the Animal Review Board at Guangdong Provincial Hospital of Chinese Medicine (approval number: 2022051). All rats were adapted in a fixed environment for 1 week (six per cage). The room temperature was maintained at 25˚C ± 2˚C, the relative humidity was 50% ± 10%, and the light–dark cycle was 12:12 h. All experimental protocols were approved by the Institutional Animal Ethics Committee of Guangdong Provincial Hospital of Traditional Chinese Medicine. All the rats fasted with free access to water for 12 h. Before the experiment, 9 rats were randomly divided into control group, medium dose OS decoction group (6.25g/g) and high dose OS decoction group (12.5g/g). 3 rats were treated with OS decoction (6.25g/g, body weight), 3 rats were treated with OS decoction (12.5g/g, body weight), and 3 rats were treated with ultra-pure water to obtain blank serum samples. Rats in each group were treated with ultra-pure water and drugs for 7 days, twice a day, and they were dissected and killed 2 hours after administration.

## 2.4 Preparation of serum

Blood was drawn from the abdominal aorta, after which serum was extracted using a procoagulant tube. The samples were left to stand for two hours after clotting had occurred, and then subsequently centrifuged at 4˚C, 3000 rpm for 15 minutes. Post centrifugation, the upper layer of clear liquid was segregated as serum. Sample serums from each group of rats were combined and preserved at -80˚C. For analysis, 1.5 ml of mixed serum was obtained from an even contribution of 250 μl from each of the six decoction-treated rats. Next, 6 ml of methanol were mixed into the serum, with the process conducted on ice. This was followed by a vortexing process for three minutes, and a centrifugation exercise at 12,000 rpm for 10 minutes. Subsequently, 7 ml of the supernatant was transferred to a clean 15 ml centrifuge tube to undergo vacuum centrifugation at 4˚C, yielding a dried sample. The dried residue was then re-dissolved in 200 μl of methanol water solution in a 1:1 (v:v) ratio, passed through a pinhole filter membrane, subsequently transferred to an injection vial lined with an inner liner and finally sent for UHPLC–Q Exactive Orbitrap–HRMS analysis.

## 2.5 Preparation of renal tissue

Upon euthanizing and dissecting the rats, two kidney tissue samples were collected. One sample was rinsed and dried with saline, subsequently placed in a cryogenic storage tube, and stored in liquid nitrogen before being transferred to a -80˚C environment for long-term conservation. The other was immediately wrapped in tin foil, flash-frozen in liquid nitrogen for 10 seconds, and then placed in a pre-cooled centrifuge tube, subsequently undergoing the same long-term storage process at -80˚C.

**UHPLC–Q exactive orbitrap–HRMS.**  For analysis, 25mg samples from each of the six rats treated with the OS decoction were dissected from the same part and mixed to achieve a

collective 150mg of kidney tissue. To this, 1.5ml of pure methanol was added along with three magnetic beads, and the mixture was homogenized using a homogenizer set at 5.65m/s for 15-seconds intervals, repeated six times with rest periods on ice. After homogenization, the solution was centrifuged at 12000rpm for 15 minutes at 4˚C. The resulting 1.5ml supernatant was then transferred to a clean 2ml centrifuge tube and vacuum dried at 4˚C. To the dried product, 150μL of 1:1 (v:v) methanol water was added, after which the mixture was filtered through a pinhole filter membrane and transferred to an injection vial with an inner liner. Finally, the sample was analyzed using UHPLC–Q Exactive Orbitrap–HRMS.

**AFADESI-MSI.** Firstly, retrieve the preserved kidney tissues, kept at -80˚C, from the refrigerator. Utilize a cryostat to obtain a significant cross-section from these tissues, which should then be sectioned into slices of 20μm thickness. Thereafter, arrange these slices on electrostatically-equipped anti-fracture microscope slides. Afterward, store these specimens in an airtight container at -20˚C for brief durations. Prior to experimental use, relocate the samples into a desiccator situated within a low-temperature setting, sustaining this environment for 15 minutes. Following this, the slices were removed and placed on a dryer at ambient room temperature, prolonging this condition until the samples thoroughly desiccate. As the final step, position the dry tissue slices onto the testing apparatus.

## 2.6 Instrumentation and experimental conditions

**UHPLC–Q exactive orbitrap–HRMS.** The experiments were performed using a Thermo Fisher Scientific UHPLC system (UltiMate 3000) coupled with a high-resolution Q Exactive Focus mass spectrometer (USA). The mass spectrometer was equipped with a heated electrospray ionization source. Chromatographic separation was performed on an BEH C18 column ($100 \times 2.1$ mm, 1.7 μm) at a flow rate of 0.2 mL/min. The injection volume was 1.0 μL, and the column temperature was 35˚C. A gradient program using a 0.1% formic acid solution in water as phase A and acetonitrile as phase D was adopted. The gradient was applied as follows: 0-2min,5% D;2-20min,5%-25% D;20-42min, 25%-95% D;42-47min,95% D;47–47.1min,95%-5% D;47.1-50min,5% D. Positive and negative ions are detected separately; the full MS scan range was m/z 120–1000, and the resolution was 70,000. The MS parameters of the positive-ion mode were as follows: the sheath gas flow rate was set at 35 L/min, the auxiliary gas flow rate was set at 10 L/min, the spray voltage of positive was 3.5 kV and the negative was -3.2 kV, the capillary temperature was 320˚C, and the auxiliary gas heater temperature was 350˚C. The AGC target was 1e6, and the maximum IT was 100 ms. The MS/MS scanning mode was a datadependent $ms^2$ scan (dd-$ms^2$) with a resolution of 17,500, and the collision energy was set to the step mode (30, 40, and 50 eV). Subsequently, the AGC target was 5e4, and the maximum IT was set at auto. The modes of positive and negative ions are the same.

**AFADESI-MSI.** Analysis was carried out in both positive and negative-ion mode on a Q Exactive mass spectrometer (Thermo Scientific) over an m/z range of 70–1,000 at a nominal mass resolution of 70,000. A mixture of acetonitrile and water (8:2, v:v) was used as the spray solvent at a flow rate of 5μL/min. The sprayer voltages were set at 4500 V in positive-ion mode and at -4500 in negative-ion mode. The extracting gas flow was 30 L/min, and the capillary temperature was 35˚C. The MSI experiments were performed by continuously scanning the tissue surface in the x direction at a constant rate of 200μm/s, with a 200μm vertical step separating the adjacent lines in the y direction.

## 2.7 Data processing and analysis

**UHPLC–Q exactive orbitrap–HRMS.** Data acquisition was executed utilizing Thermo Xcalibur software, while the extraction and processing of the MS information were conducted

using Tracefinder4.1 software. Key MS spectrometric features included retention time, exact mass (m/z), peak intensity, and fragment ions. Essential data processing involved scan alignment, peak detection, peak-list alignment, and background reduction. According to the revised methodology, UHPLC–Q Exactive Orbitrap–HRMS data was utilized as the input data for the Tracefinder4.1 software. Traditional Chinese Medicine (TCM) composition database (OTCML, Thermo Fisher Technology; TCMSP, TCMIP, and others) was used to match and identify compounds. A successful match was identified by an absolute m/z error value of less than 2ppm, coupled with both product ion fragments and isotopes satisfying the testing parameters. A set of 60 standards were employed to obtain UHPLC–Q Exactive Orbitrap–HRMS data following the same protocol. The extracted-ion chromatogram (EIC) function was used to search m/z within a range of 10 ppm, with a retention time error of ±0.1 and at least two phases in product ion fragments. If these conditions were met, it was classified as a matching substance. In conclusion, Compound Discover 3.2 software was used to analyze the samples and metabolites.

**AFADESI-MSI.**   The collected.raw files which collected from Thermo Xcalibur software were converted into.cdf format and then imported into custom-developed imaging software (MassImager, adedicated imaging software based on the C++ programming language) for ion image reconstructions and multivariate statistical analysis. After background subtraction, region-specific MS profiles were precisely extracted.

# 3. Results and discussion

## 3.1 Identification of main components of OS

The profiling of OS contents was executed using high-resolution mass spectrometry (MS). Each peak's corresponding compounds were provisionally identified with a specific identification strategy. Initially, the OS water decoction's freeze-dried powder was dissolved and subjected to extraction in various solvents. Notably, a mixture of methanol and water at a 1:1 volume ratio was chosen for extraction, demonstrating the maximum compound detection. Under these parameters, the OS water decoction's freeze-dried powder was extracted, and the resulting spectrum was identified via the TraceFinder 4.1 software identification platform using the OTCML database. Mass spectrometry data underwent processing and analysis, matching fragment information to identify compound structures. Thereafter, different chemical components were discerned based on accurate mass measurement and MS/MS spectrum analysis. The samples were tested three times at different times to check the plots and retention times, and the data were reproducible, indicating that the method has good reproducibility. (S1 Fig)

Following the preliminary identification results, 60 standards were deployed for a focused identification. This approach entailed comparing the MS/MS spectrum obtained from the standards with the retention time and fragmentation of the OS's spectrum. The same liquid chromatography mass spectrometry conditions were applied. The process included correlating the product ions to affirm the existence of the specific compound in the OS. Figs 1 and 2 illustrates Ultra High-Performance Liquid Chromatography Mass Spectrometry (UHPLC-MS) Total Ion Chromatogram (TIC)data in the positive and negative ion modes respectively. Meanwhile, the UHPLC-MS TIC spectra of the applied standards in positive and negative ion modes are represented visually in Figs 3 and 4. Among these, the compounds recognized as present by standards are enlisted in Table 1, the compounds recognized through the database are presented in Table 2, and compounds confirmed as absent are tabulated in S2 Table. Consider Rosmarinic acid (peak 16, molecular formula C18H16O8, retention time in negative ion mode 20.13) as an example. In the instance of ppm<5, the m/z 359.07681 presented in both

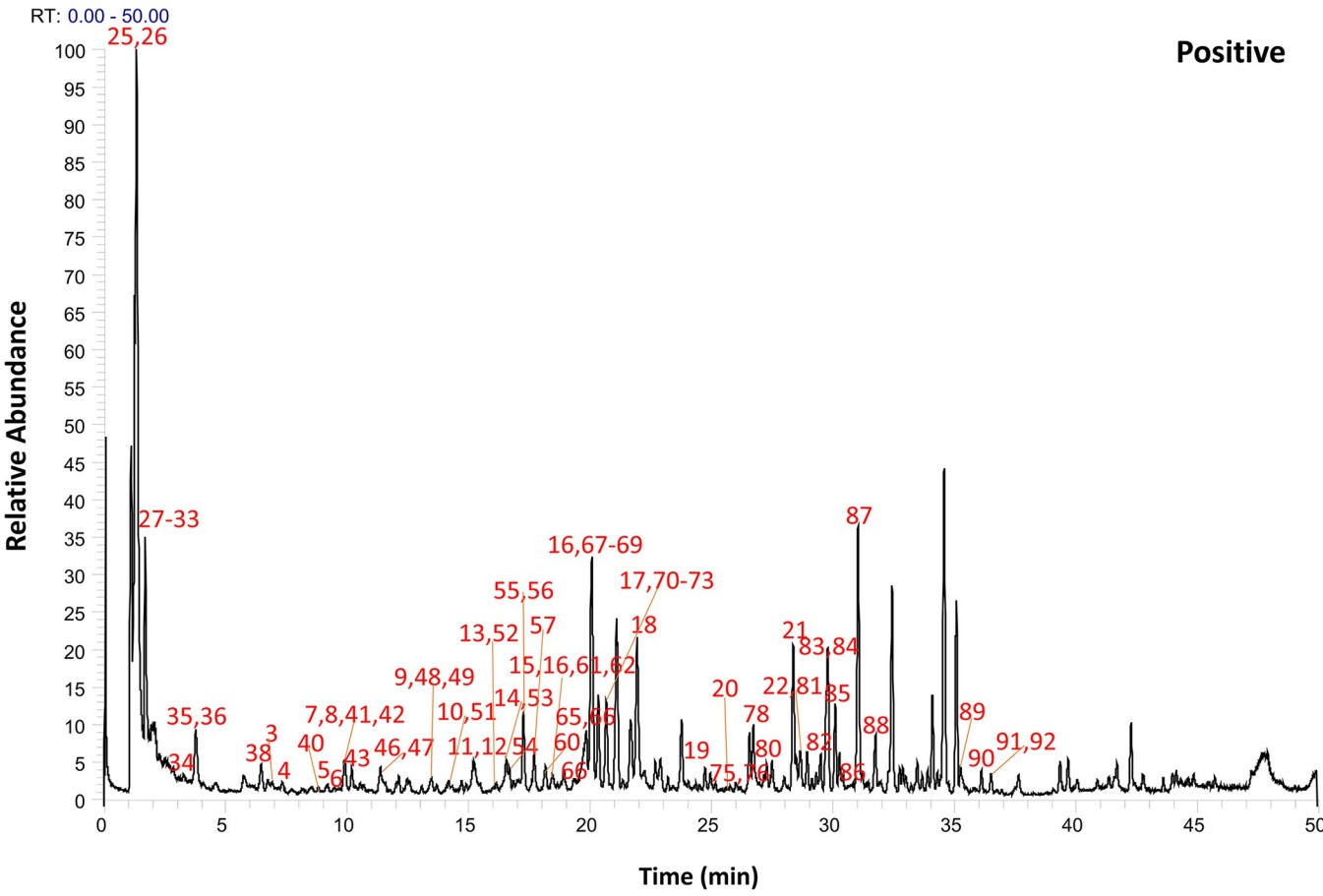

**Fig 1. The total ion flow diagram (TIC) of OS decoction in positive ion mode was obtained by UHPLC–Q Exactive Orbitrap–HRMS analysis.**

Figs 2 and 4 can produce a high peak at 20.13. Upon examining its fragment ions 161.02316, 133.02806, 136.04697, and 72.99163, these ion pairs are found in its secondary spectrum. Consequently, the presence of Rosmarinic acid in OS was confirmed. Accurate mass ions, retention times, and fragment product ions are essential throughout the compound identification process.

Employing the aforementioned identification strategy, 92 compounds were preliminarily identified, inclusive of 22 explicitly recognized by standards. These compounds primarily consist of phenylpropanoids, with flavonoids and phenolic acids being most prevalent, but also include alkaloids, terpenes, amino acids, and oligosaccharides. The variety observed aligns with previous literature. Fig 5 displays a classification of the main components according to their chemical structures. Given the varying substituents, this includes 20 flavonoids, 20 basic phenylpropanoids, 8 other phenols, 3 benzaldehyde derivatives, 10 coumarins, 4 lignans, and a single lignin, in addition to 10 terpenoids, 1 pyrrolidine alkaloid, 5 purine alkaloids, 3 amino acids, and 2 oligosaccharides. Furthermore, an additional 5 compounds were identified, inclusive of amides, phthalides, anthraquinones, and fatty acid compounds.

**3.1.1 Flavonoids.** Employing a combination of reference comparisons and TraceFinder software identification alongside standard-specific identification, 20 flavonoids were identified. Specifically, 5 flavonoids–namely Rutin, Isoquercitrin, Astragalin, Sinensetin, and Eupatorin–were distinctly identified. Broadly speaking, the mass spectrometry analysis of flavonoid

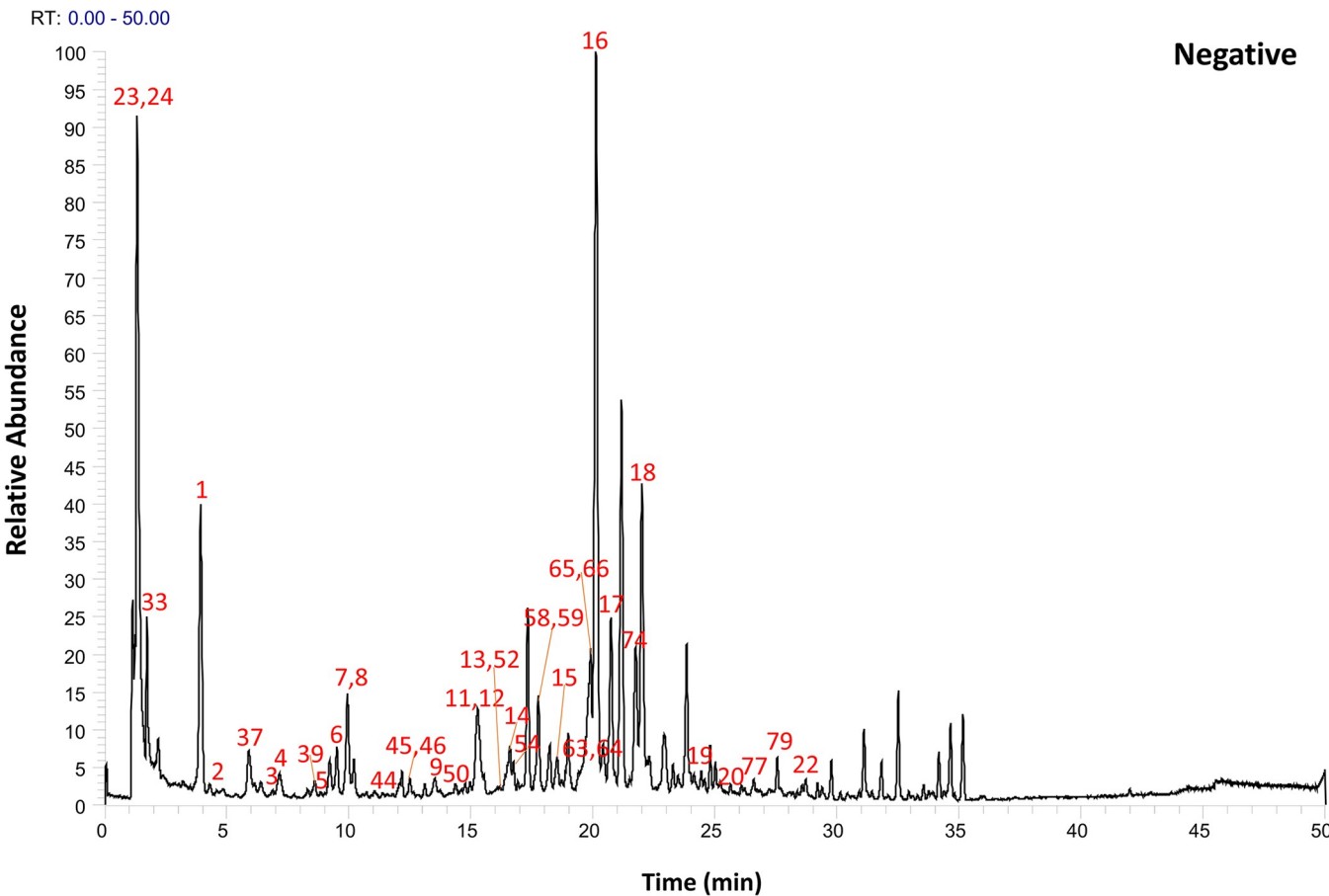

**Fig 2. The total ion flow diagram (TIC) of OS decoction in negative ion mode was obtained by UHPLC–Q Exactive Orbitrap–HRMS analysis.**

aglycones entails the loss of CH3, CO, CO2 and the fragmentation of retro-Diels-Alder (RDA) [34]. In the case of flavone glycosides, glycosidic link cleavage occurred in both positive and negative-ion modes, with 162Da (Glc), 146Da (Rha), and 308Da (rutinoside) being the characteristic neutral losses of flavonoid-O-glycosides [35]. Using Sinensetin and Rutin as exemplars, we illustrated the fragmentation patterns of these components. Peak 21 depicts the deprotonated ion of m/z 373.12820 and corresponds to the molecular formula $C_{20}H_{20}O_7$. The sub-ions of Peak 23, m/z 343.08121 $[M+H-OCH_3]^+$ and 315.08588 $[M+H-C_2H_2O_2]^-$, are products of RDA cleavage. Referencing the standard, Peak 21 was confirmed as Sinensetin (Fig 6A), while Peak 22 was identified as Eupatorin. Peak 14 exhibited a $[M-H]^-$ ion at m/z 463.08798 and a major fragment ion at 300.02713 $[M-H-C_6H_{11}O_5]^-$ due to glycoside fragment loss. Simultaneously, Peaks 13 and 14 exhibited the same fragment ions as Peak 14 at m/z 300.02713, resulting from the structural characteristics conferred by quercetin and glycoside binding. Consequently, Peaks 13 and 14 were identified as Rutin and Isoquercitrin (Fig 6B), respectively.

Through the alignment with the Traditional Chinese Medicine compound library in the TraceFinder software, 15 flavonoids were identified. These include 4'-O-Glucosylvitexin, Vicenin-2, Isovitexin, Morin, Leucoside, Galangin, 3-Hydroxyflavone, Iridin, Fisetin, Irigenin, 5-O-Demethylnobiletin, Gardenin B, 5-Hydroxy-6,7-dimethoxyflavone, Nobiletin, and 6-Demethoxytangeretin. A case in point is 4'-O-Glucosylvitexin. For instance, in negative ion

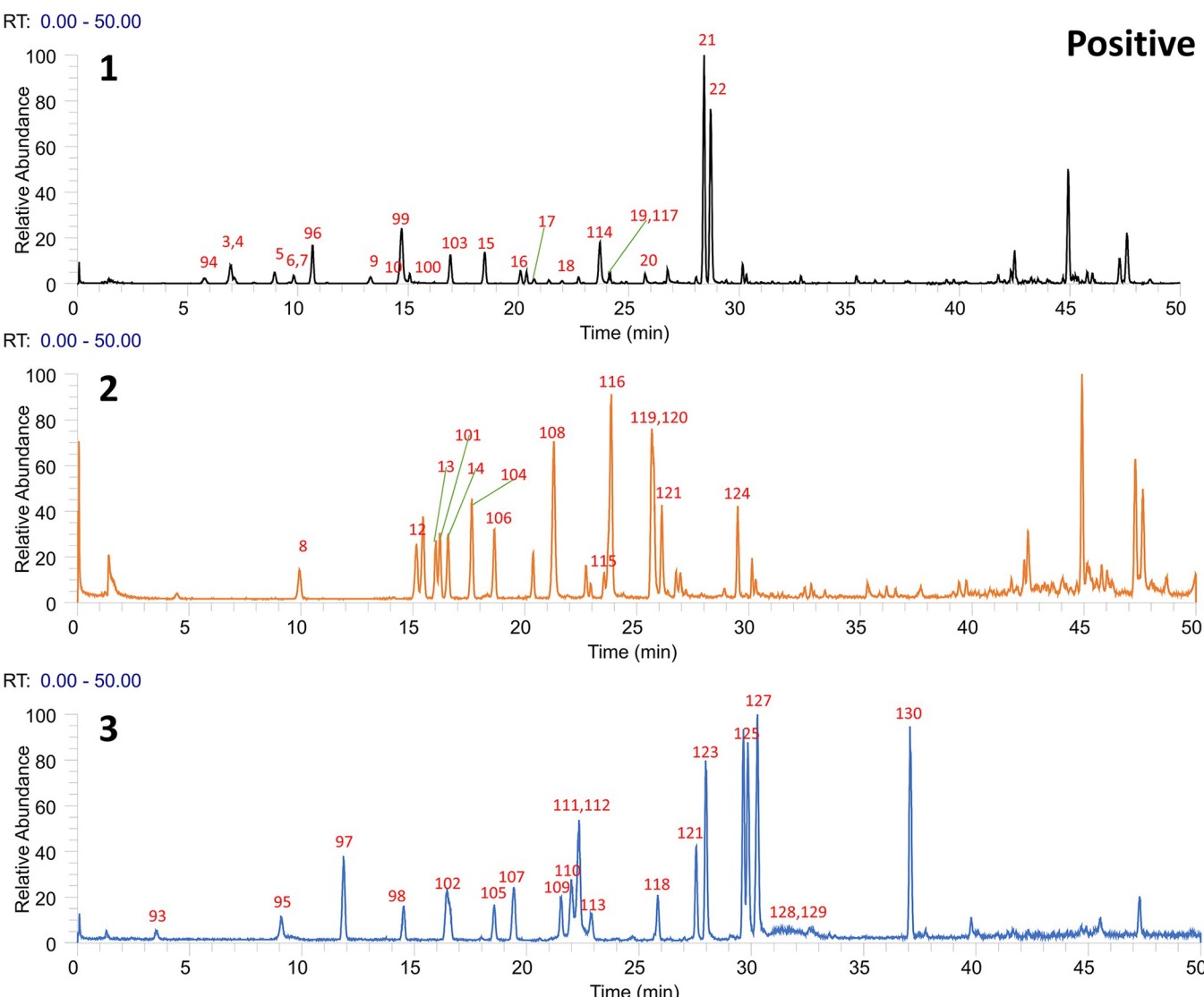

**Fig 3. The total ion flow diagram (TIC) of each standard in positive ion mode was obtained by UHPLC–Q Exactive Orbitrap–HRMS analysis.**

mode, peak 45 yields a deprotonated ion of m/z 593.15100 at 12.52, aligning with the database values. The sub-ion at m/z 413.08929 is the outcome of glycoside loss, which also matches the database. The compound is preliminarily confirmed as 4'-O-Glucosylvitexin based on its isotope identification. Other substances follow an analogous identification strategy.

**3.1.2 Phenylpropanoids.**　Following the comparison of references and TraceFinder software identification, bolstered by the specific identification of standard products, 20 phenylpropanoids were identified, of which 12 were distinctly identified. The specifically identified compounds include Danshensu, Chlorogenic acid, Cryptochlorogenic acid, Caffeic acid, cis-4-coumaric acid, Sinapic acid, ferulic acid, Rosmarinic acid, Lithospermic acid, salvianolic acid B, Salvianolic acid A, and Salvianolic acid C. These compounds exhibited identical primary fragments involving the loss of H2O and CO2 [36]. By comparison with the reference standard, Peak 8 was designated Caffeic acid, demonstrating a [M-H]⁻ ion at m/z 179.03403 and a [M-H-CO$_2$]⁻ ion at m/z 135.04385 [37]. The ion [M-H-CHO$^2$]⁻ was generated at m/z

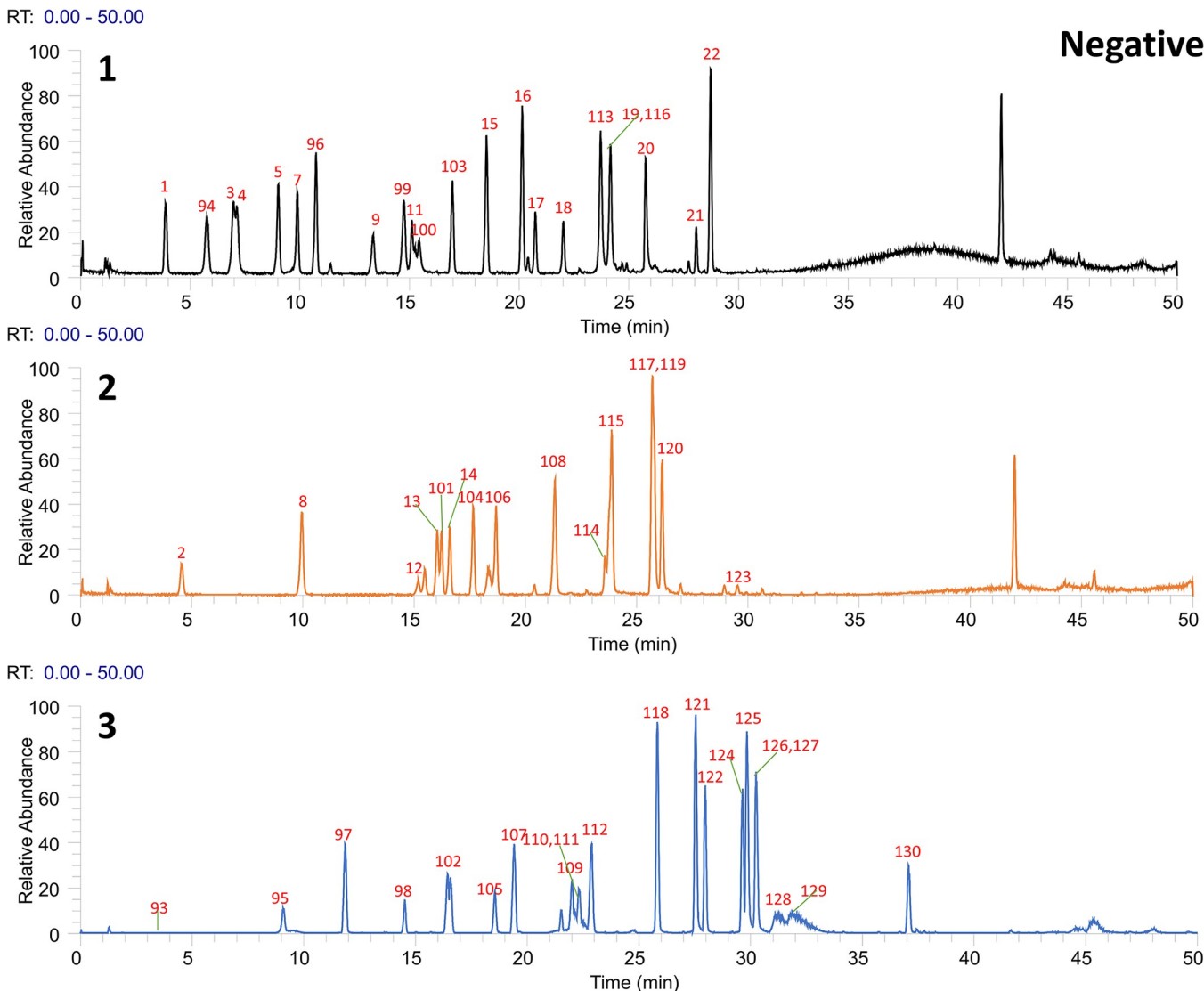

**Fig 4. The total ion flow diagram (TIC) of each standard in negative ion mode was obtained by UHPLC–Q Exactive Orbitrap–HRMS analysis.1, 2, 3 indicates that the 60 standards were randomly divided into 3 to better demonstrate and find the target compounds.**

134.03601. Following these occurrences, m/z 89.03833 was obtained by the further loss of one $CO_2$ unit, or alternatively, m/z 107.03314 was obtained by losing one CO, aligning with the bond breaking principles of phenylpropyl compounds (Fig 6D). Given that simple phenylpropanoids share the same skeleton, the fragment ions of m/z135.04385 were apparent in Peaks 1, 7, 12, 16, 17, 18, 19, and 20. However, deducing from the retention time of its reference standards, the above substances were identified as Danshensu (Fig 6C), Cryptochlorogenic acid, ferulic acid, Rosmarinic acid, Lithospermic acid, salvianolic acid B, Salvianolic acid A, and Salvianolic acid C.

The remaining eight non-standard compounds were verified solely through the Traditional Chinese Medicine database. These include p-Coumaric acid, 1-Caffeoylquinic acid, Eleutheroside B, alpha-Asarone, Pinosylvin, Calceolarioside B, Forsythoside A, and (E)-Astringin. In a similar vein, Peak 31 presents the [M+H]⁺ion of m/z 165.05463 at 1.77, with its characteristic

**Table 1. 22 compounds in OS confirmed by reference standards.**

| No. | t/min | Plausible identity | Molecular formula | Neutral mass (Da) | Pseudo molecular ion | MS1(m/z) | MS2(m/z) | ionic strength | position |
|---|---|---|---|---|---|---|---|---|---|
| 1[a,b,c] | 3.88 | Danshensu | $C_9H_{10}O_5$ | 198.170 | [M-H]- | 197.04456 | 123.04373, 134.03574, 72.99157, 135.04388 | 1.92E+08 | 1 |
| 2[a,b,c] | 4.67 | protocatechuic acid | $C_7H_6O_4$ | 154.120 | [M-H]- | 153.01800 | 108.02028, 109.02844, 91.01746, 65.00189 | 1.05E+07 | 2 |
| 3 | 6.96 | Esculin | $C_{15}H_{16}O_9$ | 340.290 | [M+H]+ | 341.08664 | 179.03398, 123.04411, 133.02850, 151.03911 | 7.45E+05 | 1 |
| | 6.95 | | | | [M-H]- | 339.07175 | 177.01820, 133.02811, 105.03323, 89.03816 | 9.36E+05 | |
| 4[a,b] | 7.13 | Protocatechualdehyde | $C_7H_6O_3$ | 138.120 | [M+H]+ | 139.03900 | 65.03880, 66.04211, 53.00262 | 8.22E+06 | 1 |
| | 7.14 | | | | [M-H]- | 137.02306 | 136.01523, 108.02023, 91.01743, 65.00185 | 4.83E+07 | |
| 5[a,b,c] | 8.96 | Chlorogenic acid | $C_{16}H_{18}O_9$ | 354.309 | [M+H]+ | 355.10217 | 163.03900, 135.04411 | 1.04E+07 | 1 |
| | 9.00 | | | | [M-H]- | 353.08746 | 191.05508, 85.02804, 93.03314, 127.03842 | 1.01E+07 | |
| 6 | 9.63 | Vanillic acid | $C_8H_8O_4$ | 168.147 | [M+H]+ | 169.04956 | 65.03880, 81.03355 | 1.97E+05 | 1 |
| 7[a,b,c] | 9.82 | Cryptochlorogenic acid | $C_{16}H_{18}O_9$ | 354.309 | [M+H]+ | 355.10217 | 163.03899, 135.04405, 117.03334, 145.02847 | 3.82E+06 | 1 |
| | 9.87 | | | | [M-H]- | 353.08746 | 135.04381, 191.05496, 93.03310, 173.04445 | 3.26E+06 | |
| 8[a,b,c] | 9.94 | Caffeic acid | $C_9H_8O_4$ | 180.157 | [M+H]+ | 181.04950 | 89.03860, 63.02318, 117.03352, 95.04917 | 2.14E+07 | 2 |
| | 9.94 | | | | [M-H]- | 179.03403 | 135.04385, 134.03601, 89.03833, 107.03314 | 1.90E+08 | |
| 9[a,b,c] | 13.30 | cis-4-coumaric acid | $C_9H_8O_3$ | 164.158 | [M+H]+ | 165.05469 | 65.03880, 91.05427, 119.04924, 75.02302 | 7.45E+05 | 1 |
| | 13.34 | | | | [M-H]- | 163.03886 | 119.04877, 93.03309, 117.03316, 65.03825 | 2.43E+06 | |
| 10 | 14.15 | Sinapic acid | $C_{11}H_{12}O_5$ | 224.210 | [M+H]+ | 225.07587 | 91.05424, 65.03876, 89.03857, 149.02335 | 3.15E+05 | 1 |
| 11[b,c] | 15.17 | ferulic acid | $C_{10}H_{10}O_4$ | 194.184 | [M+H]+ | 195.06520 | 89.03863, 78.04649, 117.03355, 63.02320 | 1.50E+06 | 2 |
| | 15.18 | | | | [M-H]- | 193.04971 | 178.02602, 134.03598 | 5.56E+05 | |
| 12[a,b] | 15.25 | Chicoric acid | $C_{22}H_{18}O_{12}$ | 474.371 | [M-H]- | 473.07224 | 179.03381, 149.00784, 135.04381, 87.00733 | 8.51E+07 | 1 |
| 13[a,b] | 16.03 | Rutin | $C_{27}H_{30}O_{16}$ | 610.518 | [M+H]+ | 611.16030 | 303.04962, 85.02843, 71.04920 | 9.47E+05 | 2 |
| | 16.03 | | | | [M-H]- | 609.14557 | 300.02713, 271.02457, 151.00223 | 1.40E+06 | |
| 14[a] | 16.57 | Isoquercitrin | $C_{21}H_{20}O_{12}$ | 464.376 | [M+H]+ | 465.10278 | 303.04950, 85.02837, 97.02835, 69.03359 | 5.53E+06 | 2 |
| | 16.59 | | | | [M-H]- | 463.08798 | 300.02713, 271.02444, 255.02917, 151.00244 | 6.60E+06 | |
| 15[a] | 18.43 | Astragalin | $C_{21}H_{20}O_{11}$ | 448.377 | [M+H]+ | 449.10779 | 287.05490, 85.02847, 69.03364, 288.05817 | 1.05E+07 | 1 |
| | 18.52 | | | | [M-H]- | 447.09290 | 284.03223, 255.02927, 277.03407, 285.03836 | 1.33E+07 | |
| 16[a,b] | 20.10 | Rosmarinic acid | $C_{18}H_{16}O_8$ | 360.315 | [M+H]+ | 361.09155 | 163.03899, 181.04948, 135.04404 | 2.53E+07 | 1 |
| | 20.13 | | | | [M-H]- | 359.07681 | 161.02316, 133.02806, 136.04697, 72.99163 | 6.53E+08 | |

(*Continued*)

**Table 1.** (Continued)

| No. | t/min | Plausible identity | Molecular formula | Neutral mass (Da) | Pseudo molecular ion | MS1(m/z) | MS2(m/z) | ionic strength | position |
|---|---|---|---|---|---|---|---|---|---|
| 17[a,b] | 20.72 | Lithospermic acid | $C_{27}H_{22}O_{12}$ | 538.456 | [M+H]+ | 539.11871 | 295.06009, 137.02339, 249.05446, 277.04941 | 1.69E+07 | 1 |
| | 20.74 | | | | [M-H]- | 537.10327 | 295.06094, 185.02312, 109.02800, 135.04370 | 1.29E+08 | |
| 18[a,b] | 21.99 | salvianolic acid B | $C_{36}H_{30}O_{16}$ | 718.614 | [M+H]+ | 719.15948 | 521.10791, 539.11847, 154.99016, 231.02867 | 1.30E+07 | 1 |
| | 22.02 | | | | [M-H]- | 717.14557 | 519.09235, 321.03992, 339.05048, 295.06055 | 4.48E+08 | |
| 19[a] | 24.25 | Salvianolic acid A | $C_{26}H_{22}O_{10}$ | 494.447 | [M+H]+ | 495.12891 | 297.07568, 331.04468, 287.05490, 154.99025 | 2.06E+05 | 1 |
| | 24.29 | | | | [M-H]- | 493.11417 | 295.06073, 162.83788, 109.02807, 164.83788 | 2.54E+07 | |
| 20[b,c] | 25.76 | Salvianolic acid C | $C_{26}H_{20}O_{10}$ | 492.431 | [M+H]+ | 493.11276 | 295.06006, 313.07040, 271.05997 | 2.12E+05 | 1 |
| | 25.75 | | | | [M-H]- | 491.09799 | 293.04523, 135.04376, 265.05005, 72.99163 | 1.74E+06 | |
| 21[a,b,c] | 28.42 | Sinensetin | $C_{20}H_{20}O_7$ | 372.369 | [M+H]+ | 373.12820 | 343.08121, 315.08588, 171.02890, 357.09689 | 3.04E+08 | 1 |
| 22[a,b] | 28.71 | Eupatorin | $C_{18}H_{16}O_7$ | 344.315 | [M+H]+ | 345.09671 | 284.06793, 312.06274, 330.07327, 108.02065 | 6.22E+07 | 1 |
| | 28.74 | | | | [M-H]- | 343.08185 | 298.01151, 270.01666, 313.03506, 285.04004 | 1.41E+07 | |

[a] represents the compound found in the methanol extract of OS powder

[b] represents the compound found in the drug-containing serum, and

[c] represents the compound found in the drug-containing kidney tissue homogenate

ion resulting in the $[M+H-C_2H_2O_3]^+$ ion at m/z 91.05429, aligning with the database information. Thus, Peak 31 was preliminarily identified as p-Coumaric acid (Fig 6E).

**3.1.3 Phenols.** Excluding flavonoids, simple phenylpropanoids, coumarins, phenylpropanals, lignans and lignins, Nine phenolic compounds were identified. These include Protocatechuic acid, Protocatechualdehyde, Vanillic acid, 6-Gingerol, 4-Methoxysalicylic acid, Caftaric acid, Ethyl caffeate, 5-Acetylsalicylic acid, and 6-Shogaol. The fundamental structure of such compounds is a benzene ring, augmented with one or more hydroxyl groups, thereby primarily involving the same $H_2O$ and $CO_2$ primary fragments [38]. For instance, Peak 2 typifies the $[M-H]^-$ characteristic ion of m/z 153.01800 occurring at 4.67. Its characteristic ion is $[M-H-CO_2]^-$ which appears in m/z 110.03168. After losing two H ions, the characteristic ion of m/z 108.02028 results. In the ionization process, one OH was lost yielding the characteristic ion of m/z 91.01746, and, following a loss of CO, the characteristic ion of m/z 65.00189 is generated. The entire ionization process primarily involved the basic fragments of $H_2O$ and $CO_2$. The compound was provisionally identified as Protocatechuic acid and confirmed through the reference standard (Fig 6F).

**3.1.4 Benzaldehyde derivatives.** Three benzaldehyde derivative compounds were identified, specifically Anisic aldehyde, Asarylaldehyde, and 2-Hydroxy-4-methoxybenzaldehyde. Based on the parent ions, fragment ions, and isotopes in the database, Peak 42 was determined to be Anisic aldehyde, Peak 43 as Asarylaldehyde, and Peak 47 as 2-Hydroxy-4-methoxybenzaldehyde. Peak 42 exhibits the $[M+H]^+$ characteristic ion at m/z 153.01800, Peak 43 presents the $[M+H]^+$ characteristic ion at m/z 197.08095, and Peak 47 showcases the $[M+H]^+$

**Table 2. 70 compounds identified in OS by database.**

| No. | t/min | Plausible identity | Molecular formula | Neutral mass (Da) | Pseudo molecular ion | Error/ ppm | MS1(m/z) | MS2(m/z) | ionic strength |
|---|---|---|---|---|---|---|---|---|---|
| 23[a,b,c] | 1.27 | Stachyose | $C_{24}H_{42}O_{21}$ | 666.578 | [M-H]- | -0.98 | 665.21747 | 383.11392, 179.05492 | 4.37E+06 |
| 24[c] | 1.29 | Raffinose | $C_{18}H_{32}O_{16}$ | 504.437 | [M+HCO2]- | -0.56 | 549.16632 | 221.02940, 179.05492, 161.04433 | 2.55E+07 |
| 25[a,b] | 1.37 | Stachydrine | $C_7H_{13}NO_2$ | 143.180 | [M+H]+ | -0.29 | 144.10194 | 58.06546, 84.08081, 144.10219, 102.05511 | 2.18E+07 |
| 26[a,b,c] | 1.38 | Adenine | $C_5H_5N_5$ | 135.127 | [M+H]+ | 0.09 | 136.06181 | 92.02435, 119.03528, 82.04028, 109.05140 | 2.30E+07 |
| 27[b] | 1.69 | Nicotinamide | $C_6H_6N_2O$ | 122.125 | [M+H]+ | -0.12 | 123.05527 | 80.04955, 96.04440, 77.00842, 95.01896 | 3.72E+06 |
| 28[a,b,c] | 1.72 | Adenosine | $C_{10}H_{13}N_5O_4$ | 267.241 | [M+H]+ | -0.44 | 268.10400 | 136.06184, 85.02830, 119.03526, 94.04000 | 9.71E+07 |
| 29 | 1.76 | Cordycepin | $C_{10}H_{13}N_5O_3$ | 251.242 | [M+H]+ | -0.32 | 252.10904 | 136.06131, 99.06000 | 4.00E+06 |
| 30[a,b,c] | 1.77 | L-Tyrosine | $C_9H_{11}NO_3$ | 181.189 | [M+H]+ | 0.29 | 182.08122 | 123.09163, 136.07607, 91.05428, 119.04922 | 1.53E+07 |
| 31[a,b] | 1.77 | p-Coumaric acid | $C_9H_8O_3$ | 164.158 | [M+H]+ | 0.22 | 165.05463 | 91.05429, 95.04917, 96.04441, 80.04950 | 3.29E+06 |
| 32[a,b,c] | 1.79 | Guanine | $C_5H_5N_5O$ | 151.126 | [M+H]+ | 0.38 | 152.05669 | 110.03495, 135.03038, 55.02945, 153.04071 | 1.49E+07 |
| 33[a,c] | 1.78 | Guanosine | $C_{10}H_{13}N_5O_5$ | 283.241 | [M+H]+ | -0.36 | 284.09888 | 110.03489, 135.03017, 152.05672, 153.04077 | 1.56E+07 |
| | 1.80 | | | | [M-H]- | -0.46 | 282.08389 | 133.01425, 150.04086, 126.02946, 108.01888 | 5.73E+06 |
| 34[a,b,c] | 3.29 | L-Phenylalanine | $C_9H_{11}NO_2$ | 165.189 | [M+H]+ | 0.54 | 166.08636 | 120.08089, 103.05428, 95.04918, 91.05426 | 1.67E+07 |
| 35 | 3.68 | 4-Methoxysalicylic acid | $C_8H_8O_4$ | 168.147 | [M+H]+ | 0.30 | 169.04961 | 55.93477, 95.04919, 77.03865, 116.96620 | 2.30E+06 |
| 36 | 3.80 | Daphnetin | $C_9H_6O_4$ | 178.140 | [M+H]+ | 0.00 | 179.03394 | 123.04369, 95.04919, 105.03362, 77.03864 | 8.91E+06 |
| 37[a,b] | 5.88 | Caftaric acid | $C_{13}H_{12}O_9$ | 312.229 | [M-H]- | -0.58 | 311.04065 | 149.00789, 179.03456, 135.04390 | 5.28E+07 |
| 38[a,b,c] | 6.50 | L(-)-Tryptophan | $C_{11}H_{12}N_2O_2$ | 204.230 | [M+H]+ | -0.21 | 205.09717 | 118.06528, 146.06007, 91.05430, 115.05431 | 2.01E+07 |
| 39[b,c] | 8.85 | 1-Caffeoylquinic acid | $C_{16}H_{18}O_9$ | 354.309 | [M-H]- | -0.03 | 353.08786 | 191.03378, 205.04974, 233.04494, 163.03886 | 1.01E+07 |
| 40[b,c] | 8.85 | scopolin | $C_{16}H_{18}O_9$ | 354.309 | [M+H]+ | -0.96 | 355.10236 | 193.04996, 205.04968, 235.06013, 219.06534 | 1.04E+07 |
| 41[b] | 9.48 | Eleutheroside B | $C_{17}H_{24}O_9$ | 372.367 | [M+NH4]+ | -0.20 | 390.17584 | 161.05984, 105.06996, 133.064888, 70.06525 | 4.75E+05 |
| 42 | 9.71 | Anisic aldehyde | $C_8H_8O_2$ | 136.148 | [M+H]+ | 0.32 | 137.05984 | 94.04140, 55.93478, 65.03882, 114.97095 | 1.27E+06 |
| 43 | 10.33 | Asarylaldehyde | $C_{10}H_{12}O_4$ | 196.200 | [M+H]+ | 0.55 | 197.08095 | 151.03896, 95.04921 | 5.60E+05 |
| 44[b,c] | 11.31 | Gentiopicroside | $C_{16}H_{20}O_9$ | 356.325 | [M-H]- | 0.59 | 355.10355 | 149.05965, 134.03601, 178.02599, 193.05020 | 1.02E+06 |
| 45[b] | 12.52 | 4'-O-Glucosylvitexin | $C_{27}H_{30}O_{15}$ | 594.518 | [M-H]- | 0.81 | 593.15100 | 413.08929, 353.06628, 383.07690, 473.10809 | 1.04E+07 |
| 46[a,b] | 12.52 | Vicenin-2 | $C_{27}H_{30}O_{15}$ | 594.518 | [M+H]+ | -0.27 | 595.16547 | 85.02842, 232.07091, 70.06526, 233.07614 | 1.14E+07 |
| | 12.52 | | | | [M-H]- | 0.81 | 593.15118 | 383.07690, 473.10828, 593.15033, 353.06635 | 1.04E+07 |
| 47 | 12.54 | 2-Hydroxy-4-methoxybenzaldehyde | $C_8H_8O_3$ | 152.150 | [M+H]+ | 0.04 | 153.05466 | 110.03636, 65.03877, 116.96615, 106.96291 | 1.05E+06 |

(*Continued*)

**Table 2.** (Continued)

| No. | t/min | Plausible identity | Molecular formula | Neutral mass (Da) | Pseudo molecular ion | Error/ ppm | MS1(m/z) | MS2(m/z) | ionic strength |
|---|---|---|---|---|---|---|---|---|---|
| 48[b] | 13.52 | 3-n-Butylphathlide | $C_{12}H_{14}O_2$ | 190.240 | [M+H]+ | -0.01 | 191.10666 | 110.03636, 91.05427, 79.05428, 116.96619 | 2.10E+06 |
| 49[b] | 13.52 | alpha-Asarone | $C_{12}H_{16}O_3$ | 208.254 | [M+H]+ | 0.14 | 209.11734 | 121.06508, 79.05428, 91.05427, 95.04920 | 6.36E+06 |
| 50 | 14.36 | Isoimperatorin | $C_{16}H_{14}O_4$ | 270.280 | [M-H]- | -0.53 | 269.08182 | 121.02802, 109.02812, 159.04384 | 8.66E+06 |
| 51 | 14.58 | Pinosylvin | $C_{14}H_{12}O_2$ | 212.244 | [M+H]+ | -0.17 | 213.09109 | 167.09000 | 5.74E+05 |
| 52[c] | 16.04 | Isovitexin | $C_{21}H_{20}O_{10}$ | 432.378 | [M+H]+ | -0.08 | 433.11328 | 313.07059, 283.06009, 284.06689, 297.07559 | 9.54E+05 |
| | 16.04 | | | | [M-H]- | -0.59 | 431.09814 | 283.06097, 311.05582, 135.04382, 179.03383 | 5.59E+05 |
| 53[a] | 16.59 | Morin | $C_{15}H_{10}O_7$ | 302.236 | [M+H]+ | -1.12 | 303.04965 | 229.04941, 137.02354, 153.01831, 68.99724 | 2.29E+06 |
| 54[a] | 16.74 | Leucoside | $C_{26}H_{28}O_{15}$ | 580.492 | [M+H]+ | -0.34 | 581.15027 | 287.05502, 293.08026 | 3.07E+06 |
| | 16.74 | | | | [M-H]- | -0.18 | 579.13477 | 284.03223, 285.03979, 255.02966, 286.04358 | 3.37E+06 |
| 55 | 17.22 | Galangin | $C_{15}H_{10}O_5$ | 270.237 | [M+H]+ | -0.83 | 271.05997 | 153.01828, 91.05428, 119.04926, 68.99725 | 3.81E+05 |
| 56 | 17.31 | 3-Hydroxyflavone | $C_{15}H_{10}O_3$ | 238.238 | [M+H]+ | -1.48 | 239.07007 | 165.06996, 139.05424, 115.05430, 168.05690 | 1.12E+07 |
| 57[a] | 17.74 | Skimmin | $C_{15}H_{16}O_8$ | 324.280 | [M+H]+ | -1.39 | 325.09174 | 163.03900, 89.02861, 135.04410, 117.03355 | 2.93E+06 |
| 58 | 17.75 | Calceolarioside B | $C_{23}H_{26}O_{11}$ | 478.446 | [M-H]- | 0.06 | 477.13956 | 161.02319, 135.04381, 109.02813, 185.02344 | 7.38E+05 |
| 59[a] | 17.76 | Iridin | $C_{24}H_{26}O_{13}$ | 522.455 | [M-H]- | -0.46 | 521.12988 | 359.09866, 161.02319, 179.03391, 197.04449 | 1.53E+08 |
| 60 | 18.00 | Ethyl caffeate | $C_{11}H_{12}O_4$ | 208.211 | [M+H]+ | -0.51 | 209.08093 | 117.03346, 91.05426, 95.04916, 89.03859 | 1.14E+06 |
| 61 | 18.46 | 5,7-Dihydroxy-4-methylcoumarin | $C_{10}H_8O_4$ | 192.168 | [M+H]+ | 0.50 | 193.04964 | 103.05439, 107.04922, 103.05439, 91.05430 | 1.14E+06 |
| 62[b] | 18.51 | Fisetin | $C_{15}H_{10}O_6$ | 286.236 | [M+H]+ | -1.23 | 287.05496 | 137.02341, 241.04860, 153.01834, 121.02853 | 5.08E+06 |
| 63[c] | 19.14 | Tracheloside | $C_{27}H_{34}O_{12}$ | 550.552 | [M+HCO2]- | -0.07 | 595.20282 | 287.14465, 369.13428, 339.08694, 135.04375 | 4.05E+05 |
| 64 | 19.21 | Forsythoside A | $C_{29}H_{36}O_{15}$ | 624.587 | M-H | -0.62 | 623.19812 | 161.02328, 135.04375, 369.13428, 339.08694 | 4.38E+05 |
| 65[a] | 19.42 | (-)-Pinoresinol-4-O-glucoside | $C_{26}H_{32}O_{11}$ | 520.530 | M+H | -0.03 | 521.10822 | 175.07596, 127.05988, 139.03909, 249.05476 | 9.81E+05 |
| | 19.45 | | | | M-H | -0.68 | 519.18683 | 357.13000, 151.04000 | 2.02E+05 |
| 66 | 19.48 | Aurantio-obtusin-beta-D-glucoside | $C_{23}H_{24}O_{12}$ | 492.429 | M+H | -1.12 | 493.13422 | 331.08127, 316.05777 | 1.68E+06 |
| | 19.48 | | | | M-H | -1.04 | 491.11932 | 313.04233, 476.09558, 299.01920, 314.04233 | 4.86E+05 |
| 67[a] | 20.12 | 7-Hydroxycoumarine | $C_9H_6O_3$ | 162.142 | M+H | -0.65 | 163.03897 | 107.04856, 134.03616, 145.88950, 177.87915 | 3.17E+08 |
| 68[c] | 20.12 | Irigenin | $C_{18}H_{16}O_8$ | 360.315 | M+H | -1.50 | 361.09149 | 119.04949, 163.03900, 135.04402, 222.06310 | 2.53E+07 |
| 69[a] | 20.13 | 5-Acetylsalicylic acid | $C_9H_8O_4$ | 180.157 | M+H | -1.07 | 181.04945 | 107.04912, 89.03860, 63.02319, 117.03355 | 7.24E+07 |
| 70 | 20.70 | Imperatorin | $C_{16}H_{14}O_4$ | 270.280 | M+H | -0.99 | 271.09644 | 147.04410, 175.03903, 103.05425, 147.04410 | 2.60E+05 |
| 71 | 20.72 | Isopsoralen | $C_{11}H_6O_3$ | 186.163 | M+H | -0.73 | 187.03899 | 131.04922, 103.05429, 95.04918, 105.04478 | 2.85E+06 |

(*Continued*)

**Table 2.** (Continued）

| No. | t/min | Plausible identity | Molecular formula | Neutral mass (Da) | Pseudo molecular ion | Error/ ppm | MS1(m/z) | MS2(m/z) | ionic strength |
|---|---|---|---|---|---|---|---|---|---|
| 72 | 20.74 | Fraxetin | $C_{10}H_8O_5$ | 208.167 | M+H | 0.14 | 209.04454 | 149.02405, 68.99727, 65.03881, 121.02858 | 2.29E+06 |
| 73 | 20.78 | Glabrolide | $C_{30}H_{44}O_4$ | 468.668 | M+H | 0.00 | 469.33124 | 95.04922, 68.99727, 65.03881, 121.02858 | 1.28E+06 |
| 74[a,b] | 23.48 | Methyl rosmarinate | $C_{19}H_{18}O_8$ | 374.341 | M-H | 0.10 | 373.09290 | 179.03891, 135.04385, 160, 01538, 72.99165 | 2.22E+07 |
| 75[a,b] | 26.24 | (+)-Nootkatone | $C_{15}H_{22}O$ | 218.335 | M+H | -0.46 | 219.17441 | 81.07000, 105.06998, 91.05431, 79.05429 | 2.72E+06 |
| 76 | 26.39 | 5-O-Demethylnobiletin | $C_{20}H_{20}O_8$ | 388.368 | M+H | -0.90 | 389.12350 | 359.07590, 169.01320, 215.01880, 341.06561 | 5.19E+05 |
| 77 | 26.42 | (E)-Astringin | $C_{20}H_{22}O_9$ | 406.383 | M-H | -0.45 | 405.11899 | 243.06578, 108.02029, 152.01022, 135.00740 | 1.31E+06 |
| 78[a,b,c] | 26.62 | Gardenin B | $C_{19}H_{18}O_7$ | 358.342 | M+H | -1.39 | 359.11234 | 311.05341, 329.06519, 153.01839, 301.07040 | 1.18E+08 |
| 79[b] | 26.66 | Pedunculoside | $C_{36}H_{58}O_{10}$ | 650.840 | [M+HCO2]- | -0.81 | 695.40094 | 487.34274, 649.39545, 605.40576, 339.05011 | 6.33E+05 |
| 80 | 27.13 | 5-Hydroxy-6,7-dimethoxyflavone | $C_{17}H_{14}O_5$ | 298.290 | M+H | -0.71 | 299.09134 | 238.06230, 266.05731, 108.02059, 154.02608 | 4.37E+05 |
| 81 | 28.85 | Quillaic Acid | $C_{30}H_{46}O_5$ | 486.683 | M+H | -0.38 | 487.34201 | 187.14851, 119.08585, 107.08578, 133.10129 | 5.46E+05 |
| 82[a,b,c] | 29.55 | Eudesmin | $C_{22}H_{26}O_6$ | 386.438 | M+H | -1.30 | 387.18018 | 187.07204, 181.08600, 167.07040, 107.04924 | 1.07E+07 |
| 83[b] | 29.69 | Nobiletin | $C_{21}H_{22}O_8$ | 402.395 | M+H | -0.93 | 403.13895 | 373.09149, 183.02893, 211.02371, 327.08591 | 1.07E+06 |
| 84[a] | 29.86 | 6-Demethoxytangeretin | $C_{19}H_{18}O_6$ | 342.343 | M+H | -1.32 | 343.11746 | 313.07043, 285.07529, 153.01834, 299.09122 | 2.91E+08 |
| 85[b] | 30.14 | 6-Shogaol | $C_{17}H_{24}O_3$ | 276.371 | M+H | -0.48 | 277.17957 | 137.05984, 81.06996, 55.01823, 79.05428 | 3.42E+06 |
| 86[b,c] | 30.90 | Senkyunolide A | $C_{12}H_{16}O_2$ | 192.254 | M+H | 0.42 | 193.12244 | 91.05421, 107.04918, 95.04919, 135.04413 | 5.50E+05 |
| 87[a] | 31.09 | Croceic acid | $C_{20}H_{24}O_4$ | 328.402 | M+H | -1.29 | 329.17456 | 197.09666, 215.10657, 187.11154, 129.06992 | 3.93E+07 |
| 88[a] | 31.75 | Asiatic acid | $C_{30}H_{48}O_5$ | 488.699 | M+H | -0.31 | 489.35776 | 205.15895, 201.16382, 147.11690, 187.14812 | 2.15E+06 |
| 89 | 35.45 | Cryptotanshinone | $C_{19}H_{20}O_3$ | 296.360 | M+H | -0.28 | 297.14847 | 251.14305, 57.07027, 67.05444, 254.09375 | 3.43E+05 |
| 90[a,b] | 36.01 | alpha-Linolenic acid | $C_{18}H_{30}O_2$ | 278.430 | M+H | -0.39 | 279.23181 | 67.05437, 81.06988, 95.08549, 79.05421 | 3.83E+06 |
| 91 | 36.34 | Corosolic acid | $C_{30}H_{48}O_4$ | 472.700 | M+H | -0.05 | 473.36282 | 205.15848, 203.17967, 95.08569, 189.16364 | 5.85E+05 |
| 92[c] | 36.34 | Ursonic acid | $C_{30}H_{46}O_3$ | 454.684 | M+H | -0.22 | 455.35223 | 205.15863, 203.17944, 95.08552, 107.08553 | 5.56E+05 |

[a] represents the compound found in the methanol extract of OS powder

[b] represents the compound found in the drug-containing serum, and

[c] represents the compound found in the drug-containing kidney tissue homogenate

characteristic ion at m/z 153.05466. The corresponding characteristic ionizers are m/z 94.04140, m/z 151.03896, and m/z 106.96291 respectively, with their ionization fragmentation modes being the same, involving the loss of one $CO_2$ and one $H_2O$ molecule.

**3.1.5 Coumarins.** Primarily, coumarins are a category of phenylpropanoid compounds. Through a meticulous evaluation of standard products and the traditional Chinese medicine

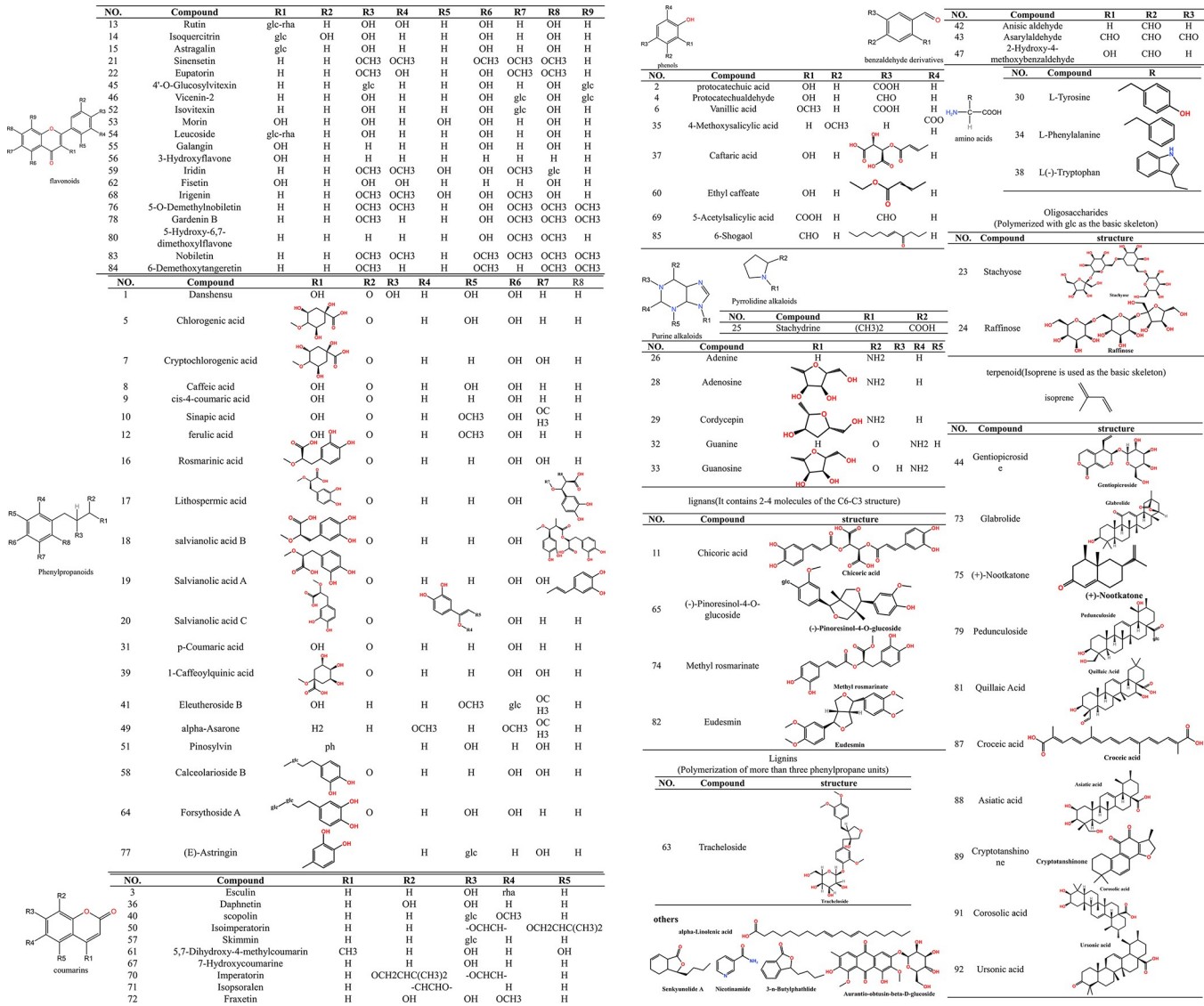

**Fig 5. Chemical structures of the main constituents identified in OS decoction.**

database in TraceFinder, 10 specific coumarins were identified, including Esculin, Daphnetin, Scopolin, Isoimperatorin, Skimmin, 5,7-Dihydroxy-4-methylcoumarin, 7-Hydroxycoumarine, Imperatorin, Isopsoralen, and Fraxetin. Notably, peak 31 represents Esculin, which has a characteristic $[M+H]^+$ ion in m/z 341.08664 due to an existing rha glucoside. The rha glucoside is preferentially lost during the fragmentation process resulting in the $[M+H-C_6H_{10}O_5]^+$ characteristic ion of m/z 179.03398. Considerably, additional loss of CO results in a $[M+-H-C_7H_{10}O_6]^+$ characteristic ion of m/z 151.03911, and further loss of one $H_2O$ molecule results in a $[M+HC_7H_{12}O_7]^+$ characteristic ion of m/z 133.02850. Alternatively, the loss of one CO molecule yields an m/z 123.04411 $[M+H-C_8H_{10}O_7]^+$ characteristic ion.

**3.1.6 Lignans and lignins.** Four lignans and one lignin compound were successfully identified; these include Chicoric acid, (-)-Pinoresinol-4-O-glucoside, Methyl rosmarinate, Eudesmin (classified as lignans), and Tracheloside (classified as lignin). The identification was confirmed by cross-checking the parent ion, fragment ion, and retention time against a

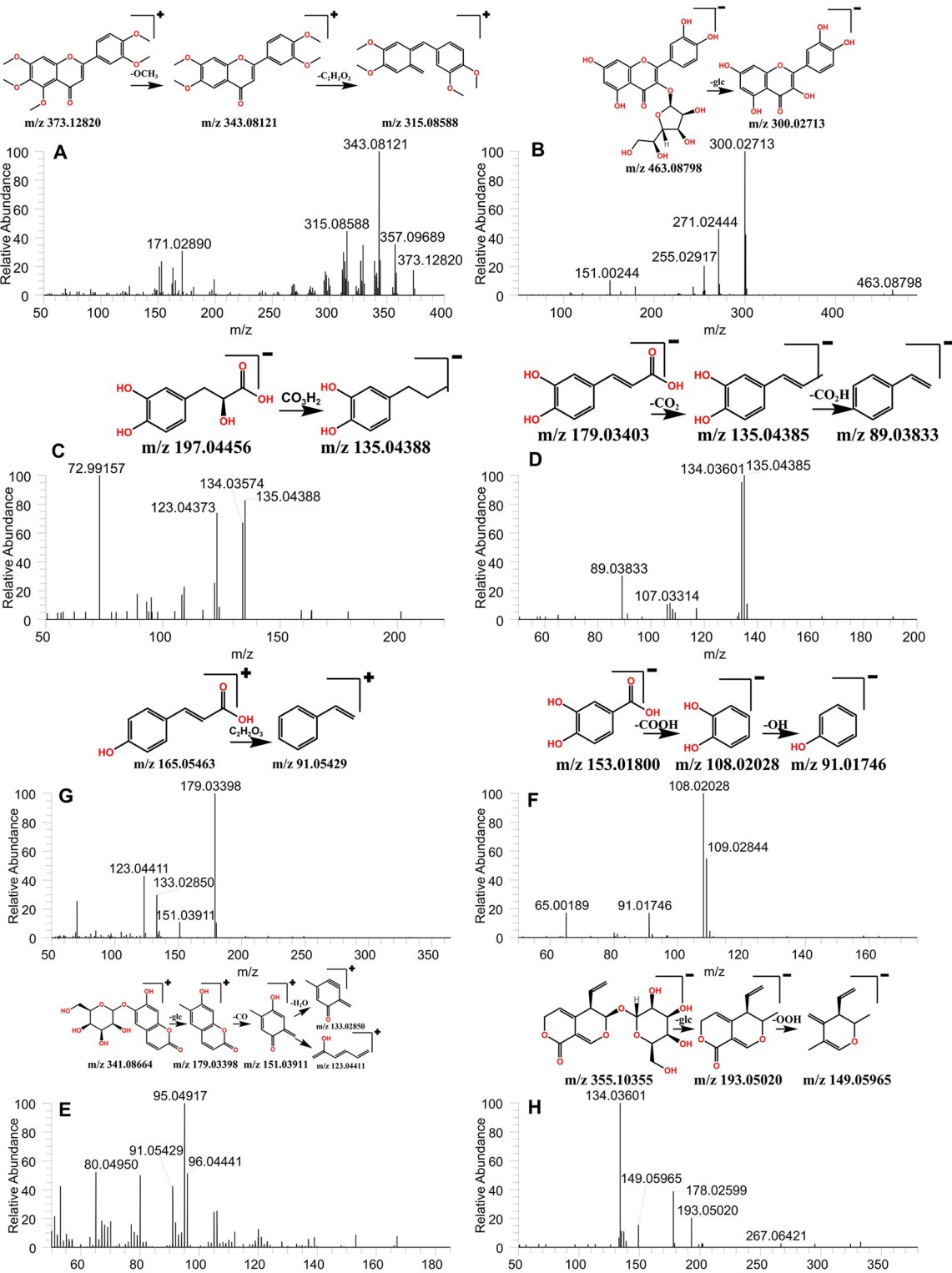

**Fig 6. Tandem mass spectra and possible fragmentation pathways of the chemical compounds.** (A) Sinensetin, (B) Isoquercitrin, (C) Danshensu, (D) Caffeic acid, (E) cis-4-coumaric acid, (F)protocatechuic acid, (G) Esculin, (H) Gentiopicroside.

standard. Consequently, peak 11 was determined as Chicoric acid. According to the TraceFinder database, peak 65 was determined to be (-)-Pinoresinol-4-O-glucoside, peak 74 as Methyl rosmarinate, peak 82 as Eudesmin, and peak 63 was Tracheloside.

**3.1.7 Terpenoids.** Eight terpenoids were identified, including Gentiopicroside, Glabrolide, (+)-Nootkatone, Pedunculoside, Quillaic Acid, Croceic acid, Asiatic acid, Cryptotanshinone, Corosolic acid, and Ursonic acid. Terpene compounds are characterized by their basic unit, isoprene, which allows them to be categorized as monoterpene, sesquiterpene, or diterpene, based on the quantity of isoprene present. The instance of peak 44 corresponds to Gentiopicroside, pertaining to the iridoid class, while peak 75 is assigned to (+)-Nootkatone, a type of sesquiterpenoids. Peaks 87 and 89 correspond to Croceic acid and Cryptotanshinone respectively, denoting diterpenoids. Glabrolide (peak 73), Pedunculoside (peak 79), Quillaic Acid (peak 81), Asiatic acid (peak 88), Corosolic acid (peak 91), and Ursonic acid (peak 92) are classified under triterpenes. Terpenoids adhere to specific rules for bond breaking under mass spectrum analysis. For example, if there are double bonds in the ring, a characteristic RDA cleavage tends to occur. Alternatively, if no double bond is present, the carbon ring often fragments into two parts. In some scenarios, both RDA cracking and carbon-ring cracking can happen in parallel. For tetracyclic triterpenoids, the commonly observed rupture usually involves the loss of a side chain [39]. Take peak 44, for instance. Here, Gentiopicroside demonstrates a characteristic $[M-H]^-$ ion at m/z 355.10355. Upon losing a glucoside and subsequent RDA cleavage of OOH, an $[M+H-C_8H_{14}O_6]^+$ ion is generated at m/z 149.05965. This set of reactions concurs with the bond-breaking norms of Gentiopicroside. Consequently, peak 44 is confirmed as Gentiopicroside, illustrated in Fig 6H.

**3.1.8 Alkaloids.** In the OS decoction, six alkaloids were identified, including Stachydrine, classified as a pyrrolidine alkaloid, and Adenine, Adenosine, Cordycepin, Guanine, and Guanosine, categorized as purine alkaloids. Taking Adenosine as an example [40, 41], the fragment ion of m/z 136.06155 $[M+H-C_5H_8O_4]^+$is produced in Adenosine, aligning with the sub-ion fragment reported in the literature, therefore we can affirmatively identify peak 28 as Adenosine. The method of identification for the remaining peaks is consistent with the above, therefore peak 25 corresponds to Stachydrine, peak 26 to Adenine, peak 29 to Cordycepin, peak 32 to Guanine and peak 33 to Guanosine.

**3.1.9 Amino acids.** Three amino acids were identified through the traditional Chinese medicine database, namely L-Tyrosine, L-Phenylalanine, and L(-)-Tryptophan. Peak 30 displayed a pseudomolecular ion at m/z 182.08127, generating major ions at m/z 136.07570 ($[M+H–COOH]^+$) and m/z 119.04934 ($[M+H–COOH–NH_3]^+$). The fragment ion at m/z 182.08127, following a neutral loss of COOH, resulted in a base peak at m/z 136.07607; this suggests the presence of a carboxyl group [42]. Furthermore, the fragment ion at m/z 136.07607, after the neutral loss of NH2 along with an H atom, yielded a base peak at m/z 119.04922 signifying the presence of an amino group [43]. Hence, peak 30 was assigned as L-Tyrosine. Similarly, peak 34 and peak 38 are tentatively classified as L-Phenylalanine and L(-)-Tryptophan, respectively.

**3.1.10 Oligosaccharides.** Stachyose and Raffinose, classified as oligosaccharides, are primarily composed of glucose. This class of compounds, known as oligosaccharide, undergoes a breakage from the O-glycosidic bond resulting in the formation of m/z 179 (glucose or fructose). The mass spectrogram reveals that peak 23 has a characteristic $[M-H]^-$ ion at m/z 665.21747 while the $[M-H-C_{18}H_{30}O_6]^-$ ion is represented at m/z 179.05492. Notably, peak 24 shows a unique $[M+HCO_2]^+$ion at m/z 549.16632 and gives rise to the $[M+HCO_2-C_{13}H_{22}O_{12}]^+$ ion appearing at m/z 179.05492 [44]. These findings correlate with the database results, thereby identifying peak 23 as Stachyose and peak 24 as Raffinose.

**3.1.11 Others.** These compounds include Nicotinamide, 3-n-Butylphathlide, Aurantio-obtusin-beta-D-glucoside, alpha-Linolenic acid, and Senkyunolide A. Aurantio-obtusin-beta-D-glucoside is classified as an anthraquinone; 3-n-Butylphathlide and Senkyunolide A fall into the phthalide category; alpha-Linolenic acid as a fatty acid; while Nicotinamide is an amide. The mass-to-charge ratio (m/z) and molecular fragments of these compounds correspond to those found in the database, enabling their preliminary identification.

## 3.2 Different compounds of methanol extract of OS decoction and methanol extract of OS powder

Due to factors such as heating and the inherent complexity of the chemical components in traditional Chinese medicine, OS production may result in new components due to complexation, hydrolysis, oxidation or reduction reactions between chemical components in the solution [45–47]. As such, identical liquid phase mass spectrometry methods were utilized to examine both the methanol extract of OS decoction and the OS powder in UHPLC–Q Exactive Orbitrap–HRMS, as portrayed in Figs 7 and 8. Observably, the same concentration of OS, post-decoction with 1:1 (v/v) methanol water extraction, manifests more peaks. Having inspected the retention time, m/z, MS/MS fragment information, structure inference, and TraceFinder database search, it was found that the compounds identified in the methanol extract of OS powder were fundamentally consistent with the methanol extract of OS decoction. However, five new compounds were identified as standard compounds namely, Esculin, Vanillic acid, Sinapic acid, ferulic acid, Salvianolic acid C, and Apigenin. Of these, Salvianolic acid B, formed by the condensation of trimolecular Danshensu and one molecular caffeic acid, possesses an ester bond that determines the instability of water-soluble components in the aqueous solution, causing it to be prone to degradation and oxidation. Initially, it can be

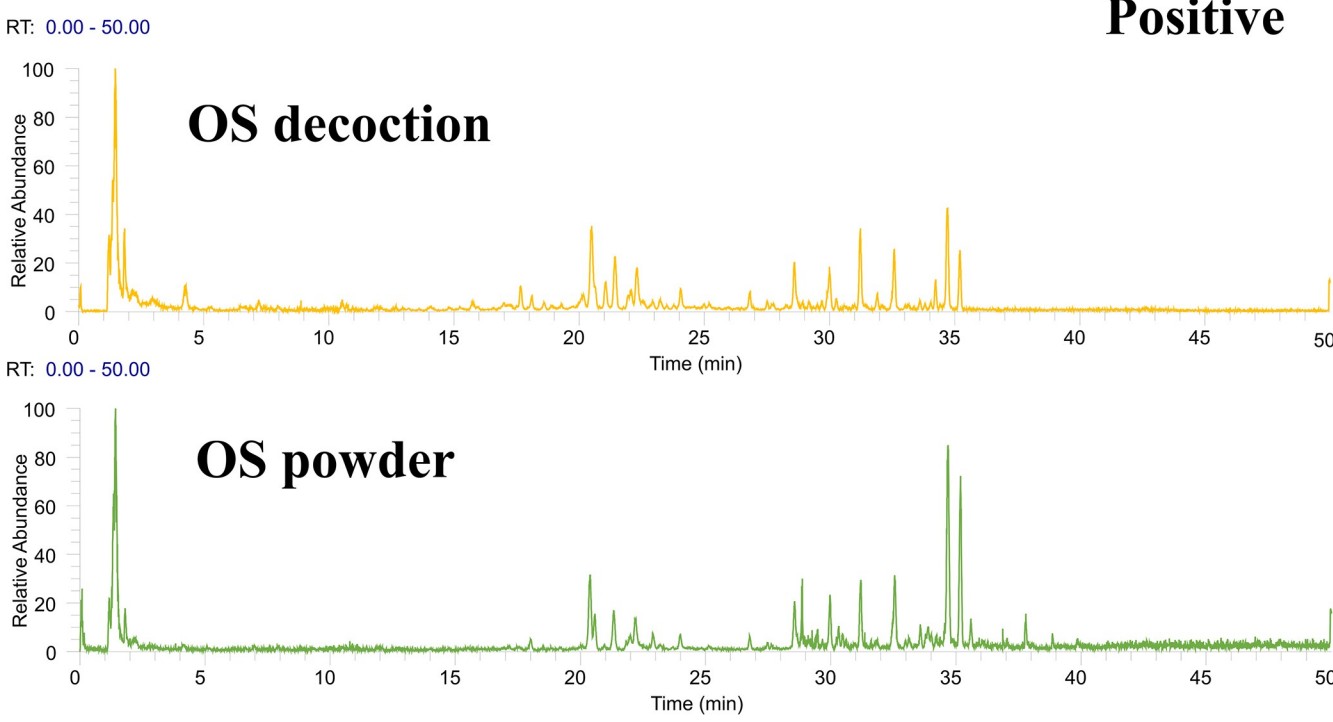

**Fig 7. TIC of OS decoction and powder obtained by UHPLC–Q Exactive Orbitrap–HRMS analysis in positive ion mode.**

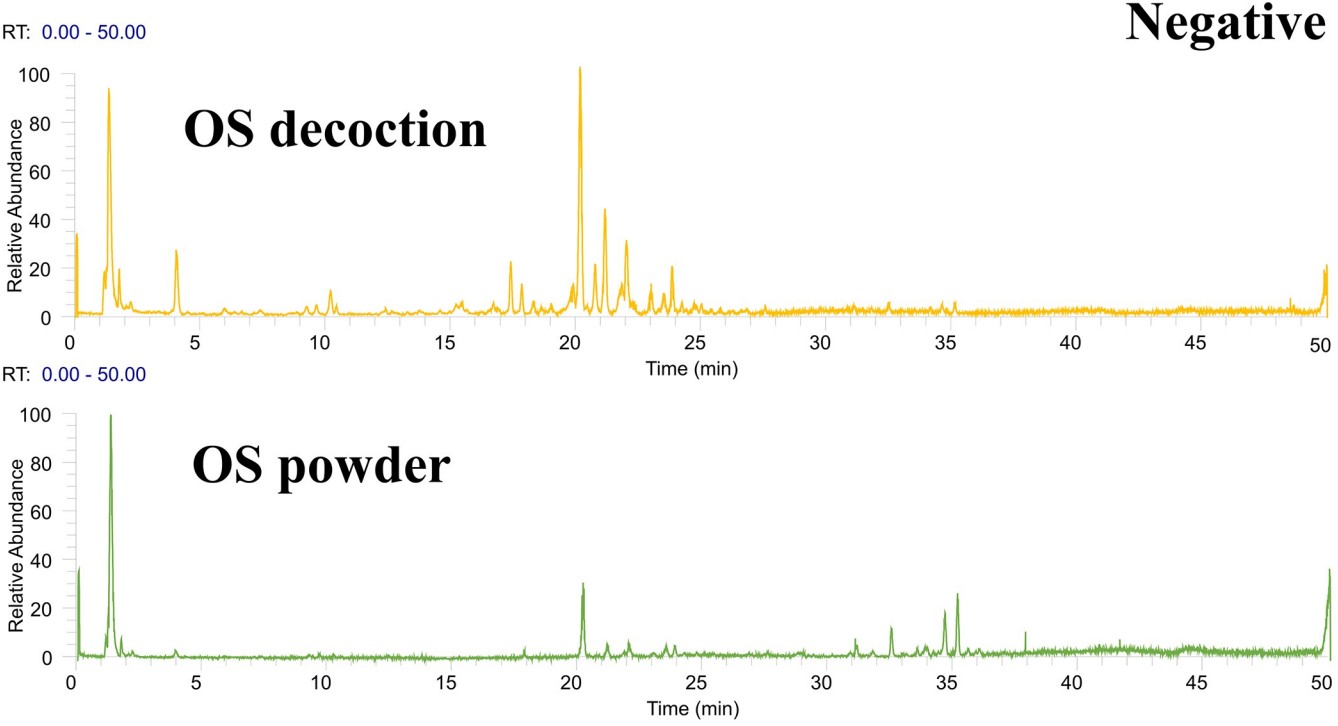

**Fig 8. TIC of OS decoction and powder obtained by UHPLC–Q Exactive Orbitrap–HRMS analysis in negative ion mode.**

metabolized into Danshensu and caffeic acid, which is further methylated into ferulic acid. Therefore, caffeic acid content in the methanol extract of the water decoction was noticeably higher than in the powder methanol extract, leading to the appearance of ferulic acid. Forty-three new compounds were identified in the methanol extract of the decoction by the Trace-Finder database. For instance, Tables 1 and 2, marked as 'a', were found in the methanol extract of OS powder. Generally, forty-eight more compounds were identified in the methanol extract of OS decoction compared to the methanol extract of OS. This analysis reflects the intricate interaction between various components caused by drug decoction, demonstrating differences between standard treatment methods of decoction and powder. However, the underlying mechanism requires further examination.

### 3.3 Screening and identification strategies of absorption prototypes and metabolites of OS in rat serum and kidney tissue

After administering an OS decoction orally for a period of seven days, both blood samples and renal tissue were collected from the abdominal aorta, specifically two hours post-administration. This two-hour interval corresponds with the half-life of the primary compounds within the OS blood, as reported in existing literature. For the purpose of determining its absorbable components, UHPLC–Q Exactive Orbitrap–HRMS screening method was initially established.

Both blank serum and renal tissue homogenate served as a negative control, while OS decoction played the role of a positive control. An extraction ion peak manifested in the OS-treated plasma, but didn't appear in the blank plasma, thereby establishing it as the original absorption component. Adhering to this standard, we identified the absorbable components of OS by comparing the accurate quality, retention time, and secondary ion fragments

corresponding to the identified elements in OS using the Thermo Xcalibur Qual Browser software. The accurate mass and tandem mass spectrometry data gleaned from the CD platform offered secondary confirmation.

Moreover, the UHPLC–Q Exactive Orbitrap–HRMS data from both the serum and blank plasma were contrasted using the CD software to identify metabolites. Notably, these metabolites appeared in the administration serum, but not the blank plasma, blank kidney tissue homogenate, or the OS decoction, and shared identical ion fragments with the prototype components.

**3.3.1 Prototype components absorbed by the blood and kidneys.** Utilizing the extraction ion chromatography (EIC) functionality of the Thermo Xcalibur Qual Browser software, we compared our findings with the identified components in the OS decoction. Our identification strategy operated under the premise that the compound's retention time error range stayed within ±0.1, the m/z error range remained below 5ppm, and the presence of the substance in the serum/kidney homogenate was confirmed when two or more identical sub-ion fragments were observed. A total of 45 prototype components were detected in the serum (S2 Fig), 16 were confirmed by standards and 29 were identified solely by the tracefinder. In contrast, only 28 prototypes were detected in the kidney (S3 Fig), with 9 being confirmed by standards and 19 identified via the tracefinder. As illustrated in Tables 1 and 2, symbols 'b' and 'c' indicate blood entry and kidney homogenate respectively, and the mean peak intensities of compounds identified in blood and kidney tisssues in control and OS groups are shown in S3 Table

**3.3.2 Analysis of metabolites of OS in rat serum.** The details of the OS prototype compounds, including their names, elemental composition, structure, and metabolic reaction transformation, were incorporated into the CD software. This allowed for the identification of metabolites through a sequence of steps, encompassing the alignment of retention time, the discovery of anticipated compounds, scoring using the Fish method, and labelling of background compounds among others. These metabolites, produced by the prototype during the initial metabolic reaction, could either be directly excreted or undergo further re-excretion through the second stage of the metabolic reaction; they can also be directly excreted by the second stage metabolic reaction alone [48, 49]. As indicated in Table 3, the transformation analysis of 16 prototypes in rat plasma through CD software yielded 49 metabolites. The prevalent phase metabolic reactions included nitro reduction, desaturation, reduction, hydration, oxidation, dehydration, and demethylation. Likewise, phase II metabolic reactions involved acetylation, methylation, and sulfonation. None of the following metabolites were detected in the controls.

Metabolism represents a biotransformation process whereby both endogenous and exogenous compounds undergo conversion into more polar products, thereby facilitating their elimination from the body. This metabolic operation consists of three distinct phases. Phase I metabolism encompasses functionalization reactions while Phase II involves a series of conjugation reactions. Phase III is denoted by transporter-mediated elimination of drugs and/or metabolites from the body, typically executed by the liver, gut, kidney, or lungs. This review disseminates elementary information on the enzymology of drug metabolism, along with elucidating potential factors that could influence the metabolic capabilities of these enzymes or modify drug responses and drug-induced toxicities.[50] None of the metabolites stemming from the compounds examined in this study were detected in blank serum. This observation suggests that these metabolites serve as prototypical compounds for OS uptake in rats, reflecting the described metabolic conditions. Given that the biological activity of phenolic compounds may transpire through the mediation of their in *vivo* metabolites [51], and considering that the key compounds of OS are predominantly phenols, it becomes imperative to investigate

Table 3. Metabolites of OS in rat serum.

| No. | t/min | Adduction | Error (ppm) | Parent compound | Transformations | MS/MS fragment | Molecular Weight | Measured mass (m/z) | Group Area: CON | Group Area: OS | Molecular formula |
|---|---|---|---|---|---|---|---|---|---|---|---|
| M1 | 1.957 | [M-H]-1 | -1.01 | Chicoric acid | Hydration | 109.02800,108.02033 | 330.05836 | 329.05109 | 0 | 6.74E+06 | $C_{13}H_{14}O_{10}$ |
| M2 | 3.009 | [M+H]+1 | 0.31 | Ferulic acid | Nitro Reduction, Oxidation, Glycine Conjugation | 119.07316,192.10210,118.06522,220.09703 | 237.10018 | 238.10746 | 0 | 6.89E+06 | $C_{12}H_{15}NO_{4}$ |
| M3 | 3.693 | [M-H]-1 | -0.36 | Chicoric acid | Hydration | 109.02811,153.05421,135.04370,108.02029 | 330.05858 | 329.05130 | 0 | 2.67E+07 | $C_{13}H_{14}O_{10}$ |
| M4 | 3.711 | [M-H]-1 | -0.86 | Cryptochlorogenic acid; Caffeic acid;Chloric acid;Chlorogenic acid; Ferulic acid | Hydration, Sulfation | 109.02811,153.05421,135.04370,108.02029 | 278.00940 | 277.00212 | 0 | 1.15E+08 | $C_{9}H_{10}O_{8}S$ |
| | | | | Rosmarinic acid | Reduction, Sulfation | | | | | | |
| | | | | Salvianolicacid B; Danshensu;Salvianolic Acid C | Sulfation | | | | | | |
| | | | | cis-4-coumaric acid | Hydration, Oxidation, Sulfation | | | | | | |
| M5 | 4.272 | [M-H]-1 | -0.39 | Chicoric acid | Reduction | 75.00739,137.05893, 71.01235 | 314.06366 | 313.05638 | 0 | 2.74E+07 | $C_{13}H_{14}O_{9}$ |
| M6 | 5.055 | [M-H]-1 | -0.28 | Lithospermic acid; Rosmarinic acid; Danshensu;Salvianolic Acid C;Salvianolicacid B | Dehydration, Glucuronide Conjugation | 135.04382,179.03384,59.01240 | 356.07425 | 355.06697 | 0 | 3.02E+07 | $C_{15}H_{16}O_{10}$ |
| | | | | Chicoric acid | Reduction, Acetylation | | | | | | |
| M7 | 5.794 | [M-H]-1 | 0.24 | Chicoric acid | Reduction | 109.02801,108.02024 | 314.06386 | 313.05658 | 0 | 3.80E+07 | $C_{13}H_{14}O_{9}$ |
| M8 | 6.019 | [M-H]-1 | -0.31 | Lithospermic acid; Rosmarinic acid; Danshensu;Salvianolic Acid C;Salvianolicacid B | Nitro Reduction, Glucuronide Conjugation | 71.01235,152.04663, 167.07022 | 344.11062 | 343.10335 | 0 | 1.54E+07 | $C_{15}H_{20}O_{9}$ |
| M9 | 7.058 | [M-H]-1 | -0.76 | Danshensu | Dehydration, Reduction, Sulfation | 123.03584,137.05946,181.04930 | 262.01452 | 261.00725 | 0 | 1.58E+08 | $C_{9}H_{10}O_{7}S$ |
| M10 | 7.254 | [M+H]+1 | 1.18 | Chlorogenic acid | Glucuronide Conjugation | 235.06103,205.04962,319.08133 | 530.12779 | 531.13507 | 0 | 2.19E+07 | $C_{22}H_{26}O_{15}$ |
| M11 | 8.074 | [M-H]-1 | -0.67 | Cryptochlorogenic acid; Caffeic acid;Chloric acid;Chlorogenic acid; Ferulic acid | Sulfation | 135.04385,134.03598,179.03366 | 259.99890 | 258.99162 | 0 | 7.33E+07 | $C_{9}H_{8}O_{7}S$ |
| | | | | Lithospermic acid; Rosmarinic acid; Danshensu;Salvianolic Acid C;Salvianolicacid B | Dehydration, Sulfation | | | | | | |

(*Continued*)

**Table 3.** (Continued)

| No. | t/min | Adduction | Error (ppm) | Parent compound | Transformations | MS/MS fragment | Molecular Weight | Measured mass (m/z) | Group Area: CON | Group Area: OS | Molecular formula |
|---|---|---|---|---|---|---|---|---|---|---|---|
| M12 | 8.362 | [M-H]-1 | -1.35 | Lithospermic acid; Rosmarinic acid; Danshensu;Salvianolic Acid C;Salvianolicacid B | Nitro Reduction, Sulfation | 122.03586,167.07022,137.05910 | 248.03512 | 247.02785 | 0 | 1.58E+07 | $C_9H_{12}O_6S$ |
| | | | | Caffeic acid | Hydration, Nitro Reduction, Sulfation | | | | | | |
| M13 | 8.459 | [M-H]-1 | -0.82 | Cryptochlorogenic acid; Caffeic acid;Chicoric acid;Chlorogenic acid;Ferulic acid;Rosmarinic acid | Reduction, Sulfation | 119.04871,137.05940,181.04948 | 262.01451 | 261.00723 | 0 | 1.37E+08 | $C_9H_{10}O_7S$ |
| | | | | Danshensu | Dehydration, Reduction, Sulfation | | | | | | |
| M14 | 8.748 | [M-H]-1 | -0.58 | Lithospermic acid; Rosmarinic acid; Danshensu;Salvianolic Acid C;Salvianolicacid B; Chicoric acid | Dehydration, Glucuronide Conjugation | 135.04382,134.03612,71.01234 | 356.07414 | 355.06686 | 0 | 4.37E+08 | $C_{15}H_{16}O_{10}$ |
| M15 | 8.888 | [M+H]+1 | 0.03 | Lithospermic acid; Rosmarinic acid; Salvianolic Acid C; Salvianolicacid B | Dehydration, Reduction | 163.03908,145.02863,135.04417,89.03850 | 180.04226 | 181.04954 | 0 | 6.03E+07 | $C_9H_8O_4$ |
| | | | | cis-4-coumaric acid | Oxidation | | | | | | |
| | | | | Danshensu; Protocatechuic acid | Dehydration | | | | | | |
| M16 | 8.891 | [M+H]+1 | 0.19 | Cryptochlorogenic acid; Rosmarinic acid;Ferulic acid;Caffeic acid; Chlorogenic acid; Chicoric acid | Dehydration | 89.03864,135.04419, 163.03905 | 162.03173 | 163.03900 | 0 | 1.39E+08 | $C_9H_6O_3$ |
| | | | | cis-4-coumaric acid | Desaturation | | | | | | |
| | | | | Protocatechuic aldehyde | Dehydration, Acetylation | | | | | | |
| | | | | Lithospermic acid; Rosmarinic acid; Danshensu;Salvianolic Acid C;Salvianolicacid B | Dehydration, Dehydration | | | | | | |

(*Continued*)

**Table 3.** (Continued)

| No. | t/min | Adduction | Error (ppm) | Parent compound | Transformations | MS/MS fragment | Molecular Weight | Measured mass (m/z) | Group Area: CON | Group Area: OS | Molecular formula |
|---|---|---|---|---|---|---|---|---|---|---|---|
| M17 | 9.201 | [M-H]-1 | -2.41 | Lithospermic acid;cis-4-coumaric acid; Salvianolic Acid C; Salvianolicacid B | Nitro Reduction, Sulfation | 93.03307,165.05472 | 246.01922 | 245.01194 | 0 | 1.03E+07 | $C_9H_{10}O_6S$ |
| | | | | Caffeic acid | Nitro Reduction, Oxidation, Sulfation | | | | | | |
| | | | | Danshensu | Desaturation, Nitro Reduction, Sulfation | | | | | | |
| M18 | 9.270 | [M-H]-1 | -1.20 | Cryptochlorogenic acid; Caffeic acid;Chicoric acid;Chlorogenic acid; Ferulic acid | Sulfation | 135.04385,134.03598,179.03375 | 259.99876 | 258.99149 | 0 | 5.63E+08 | $C_9H_8O_7S$ |
| | | | | Lithospermic acid; Rosmarinic acid; Danshensu;Salvianolic Acid C;Salvianolicacid B | Dehydration, Sulfation | | | | | | |
| M19 | 9.809 | [M-H]-1 | -0.73 | Lithospermic acid; Rosmarinic acid; Danshensu;Salvianolic Acid C;Salvianolicacid B | Dehydration, Sulfation | 135.04383,134.03592,179.03378 | 259.99888 | 258.99161 | 0 | 9.31E+07 | $C_9H_8O_7S$ |
| M20 | 10.032 | [M-H]-1 | -0.36 | Cryptochlorogenic acid Chlorogenic acid | Hydration | 59.01252,85.02793, 113.02287 | 372.10551 | 371.09824 | 0 | 1.65E+07 | $C_{16}H_{20}O_{10}$ |
| M21 | 10.608 | [M-H]-1 | -0.02 | Lithospermic acid; Rosmarinic acid; Danshensu;Salvianolic Acid C;Salvianolicacid B | Dehydration, | 59.01249,119.04882, 165.05438 | 342.09508 | 341.08780 | 0 | 3.84E+07 | $C_{15}H_{18}O_9$ |
| M22 | 10.797 | [M-H]-1 | -1.35 | Caffeic acid | Dehydration, Reduction, Sulfation | 119.04880,163.03865,93.03310 | 244.00383 | 242.99655 | 0 | 1.73E+07 | $C_9H_8O_6S$ |
| M23 | 11.067 | [M-H]-1 | -0.45 | Lithospermic acid; Rosmarinic acid; Danshensu;Salvianolic Acid C;Salvianolicacid B | Dehydration, Glucuronide Conjugation | 135.04397,134.03612,71.01233 | 356.07419 | 355.06691 | 0 | 1.79E+08 | $C_{15}H_{16}O_{10}$ |
| | | | | Chicoric acid | Reduction, Acetylation | | | | | | |
| M24 | 11.474 | [M-H]-1 | -1.93 | Cryptochlorogenic acid; Chlorogenic acid | Oxidation | 178.02597,134.03601,193.04945 | 370.08928 | 369.08200 | 0 | 2.76E+07 | $C_{16}H_{18}O_{10}$ |
| M25 | 11.562 | [M+H]+1 | 0.29 | Cryptochlorogenic acid; Rosmarinic acid;Caffeic acid;Chicoric acid; Chlorogenic acid | Dehydration, Methylation | 89.03865,63.02322, 78.04650,134.03641 | 176.04740 | 177.05467 | 0 | 2.03E+07 | $C_{10}H_8O_3$ |
| | | | | Danshensu | Dehydration, Dehydration, Methylation | | | | | | |

*(Continued)*

**Table 3.** (Continued)

| No. | t/min | Adduction | Error (ppm) | Parent compound | Transformations | MS/MS fragment | Molecular Weight | Measured mass (m/z) | Group Area: CON | Group Area: OS | Molecular formula |
|---|---|---|---|---|---|---|---|---|---|---|---|
| M26 | 11.563 | [M+K]+1 | -0.55 | Rutin | Oxidation, Methylation | 85.02832,141.01840 | 354.11602 | 393.07917 | 0 | 1.06E+07 | $C_{13}H_{22}O_{11}$ |
| M27 | 11.569 | [M+H]+1 | 0.29 | Lithospermic acid; Rosmarinic acid; Danshensu;Salvianolic Acid C;Salvianolicacid B | Dehydration, Methylation | 91.05434,65.03883, 89.03864,134.03679 | 194.05796 | 195.06524 | 0 | 7.60E+06 | $C_{10}H_{10}O_4$ |
| | | | | Caffeic acid;Chicoric acid;Chlorogenic acid | Methylation | | | | | | |
| | | | | cis-4-coumaric acid | Oxidation, Methylation | | | | | | |
| M28 | 13.064 | [M-H]-1 | -0.66 | Chicoric acid | Hydration, Nitro Reduction | 71.01237,123.04375 | 300.08432 | 299.07704 | 0 | 5.42E+07 | $C_{13}H_{16}O_8$ |
| M29 | 15.857 | [M-H]-1 | 0.44 | Rutin | Desaturation, Sulfation | 285.04019,113.02305,461.07010 | 542.03688 | 541.02960 | 0 | 7.50E+07 | $C_{21}H_{18}O_{15}S$ |
| M30 | 16.725 | [M+H]+1 | 0.02 | Ferulic acid | Hydration, Nitro Reduction, Methylation | 105.06998,107.08553,91.05429 | 196.10995 | 197.11723 | 0 | 1.86E+07 | $C_{11}H_{16}O_3$ |
| M31 | 16.919 | [M-H]-1 | -0.09 | Cryptochlorogenic acid; Chlorogenic acid | Oxidation | 193.04965,161.02313,178.02577 | 370.08996 | 369.08269 | 0 | 3.35E+07 | $C_{16}H_{18}O_{10}$ |
| M32 | 16.965 | [M+H]+1 | 0.54 | Cryptochlorogenic acid | Oxidation | 163.03903,135.04408,89.03866,85.02839 | 370.09020 | 371.09747 | 0 | 9.53E+06 | $C_{16}H_{18}O_{10}$ |
| M33 | 17.230 | [M-H]-1 | -0.79 | Chicoric acid | Nitro Reduction, Methylation | 119.04870,59.01255, 113.02267 | 296.08937 | 295.08209 | 0 | 1.87E+07 | $C_{14}H_{16}O_7$ |
| M34 | 18.393 | [M-H]-1 | -0.73 | Danshensu | Dehydration, Nitro Reduction, Glucuronide Conjugation | 134.03596,71.01228,149.05939 | 326.09993 | 325.09265 | 0 | 8.16E+06 | $C_{15}H_{18}O_8$ |
| M35 | 20.336 | [M-H]-1 | -3.63 | Cryptochlorogenic acid; Rosmarinic acid;Caffeic acid;Chicoric acid; Chlorogenic acid;Ferulic acid | Nitro Reduction, Sulfation | 148.05171,149.05954,147.04391 | 230.02406 | 229.01678 | 0 | 2.92E+07 | $C_9H_{10}O_5S$ |
| | | | | cis-4-coumaric acid | Nitro Reduction, Oxidation, Sulfation | | | | | | |
| | | | | Danshensu | Dehydration, Nitro Reduction, Sulfation | | | | | | |

*(Continued)*

**Table 3.** (Continued)

| No. | t/min | Adduction | Error (ppm) | Parent compound | Transformations | MS/MS fragment | Molecular Weight | Measured mass (m/z) | Group Area: CON | Group Area: OS | Molecular formula |
|---|---|---|---|---|---|---|---|---|---|---|---|
| M36 | 20.662 | [M+H]+1 | 0.43 | Cryptochlorogenic acid; Rosmarinic acid;Caffeic acid;Chicoric acid; Chlorogenic acid;Ferulic acid | Dehydration | 89.03860,63.02318, 95.04916,77.03874 | 162.03176 | 163.03904 | 0 | 5.89E+07 | $C_9H_6O_3$ |
| | | | | Lithospermic acid; Danshensu;Salvianolic Acid C;Salvianolicacid B | Dehydration, Dehydration | | | | | | |
| | | | | cis-4-coumaric acid | Desaturation | | | | | | |
| | | | | Protocatechuic aldehyde | Dehydration, Acetylation | | | | | | |
| M37 | 21.623 | [M+H]+1 | 0.94 | Rosmarinic acid;Caffeic acid;Chicoric acid; Chlorogenic acid; Danshensu | Dehydration, Methylation | 89.03862,63.02319, 134.03632,162.03122 | 176.04751 | 177.05479 | 0 | 1.59E+07 | $C_{10}H_8O_3$ |
| | | | | Ferulic acid | Dehydration | | | | | | |
| | | | | cis-4-coumaric acid | Desaturation, Methylation | | | | | | |
| M38 | 22.153 | [M+H]+1 | 0.99 | Sinensetin | Glucuronide Conjugation | 359.11234,329.06561,151.03873,343.08109 | 534.13787 | 535.14514 | 0 | 9.36E+06 | $C_{25}H_{26}O_{13}$ |
| M39 | 23.939 | [M+H]+1 | 0.50 | Lithospermic acid; Rosmarinic acid; Danshensu;Salvianolic Acid C;Salvianolicacid B | Nitro Reduction, Methylation | 123.08047,119.08553,67.05441,95.04914 | 182.09439 | 183.10166 | 0 | 1.52E+07 | $C_{10}H_{14}O_3$ |
| | | | | Caffeic acid | Hydration, Nitro Reduction, Methylation | | | | | | |
| | | | | Ferulic acid | Hydration, Nitro Reduction | | | | | | |
| M40 | 23.939 | [M+H]+1 | -0.12 | Ferulic acid | Nitro Reduction, Reduction, Acetylation | 91.05427,135.08049, 105.070000,149.09625 | 208.10992 | 209.11719 | 0 | 1.17E+07 | $C_{12}H_{16}O_3$ |
| M41 | 24.365 | [M+H]+1 | 0.71 | Lithospermic acid | Glucoside Conjugation | 330.07312,312.06256,256.07330 | 520.12206 | 521.12933 | 0 | 3.42E+07 | $C_{24}H_{24}O_{13}$ |
| | | | | Rosmarinic acid | Desaturation, Glucoside Conjugation | | | | | | |
| | | | | eupatorin | Glucuronide Conjugation | | | | | | |
| M42 | 24.958 | [M+H]+1 | 0.29 | (-)-Cryptochlorogenic acid | Dehydration, Methylation | 89.03864,63.02321, 95.04921,117.03360 | 176.04740 | 177.05467 | 0 | 2.03E+07 | $C_{10}H_8O_3$ |
| M43 | 25.669 | [M+H]+1 | 0.54 | Ferulic acid | Nitro Reduction, Reduction, Acetylation | 107.04919,95.04918, 135.08054,79.05427 | 208.11006 | 209.11733 | 0 | 1.06E+07 | $C_{12}H_{16}O_3$ |

(*Continued*)

**Table 3.** (Continued)

| No. | t/min | Adduction | Error (ppm) | Parent compound | Transformations | MS/MS fragment | Molecular Weight | Measured mass (m/z) | Group Area: CON | Group Area: OS | Molecular formula |
|---|---|---|---|---|---|---|---|---|---|---|---|
| M44 | 25.743 | [M-H]-1 | 0.46 | Rosmarinic acid | Nitro Reduction, Oxidation, Acetylation | 134.03595,211.06032,160.01532 | 388.11600 | 387.10872 | 0 | 1.55E+07 | $C_{20}H_{20}O_8$ |
| M45 | 25.769 | [M+H]+1 | 0.47 | Cryptochlorogenic acid | Dehydration, Methylation | 89.03867,117.03352, 63.02323,95.04923 | 176.04743 | 177.05470 | 0 | 6.62E+06 | $C_{10}H_8O_3$ |
| M46 | 27.393 | [M-H]-1 | 0.04 | Lithospermic acid; Rosmarinic acid; Danshensu;Salvianolic Acid C;Salvianolicacid B | Hydration, Palmitoyl Conjugation | 407.27985,345.27847 | 454.29307 | 453.28580 | 0 | 7.40E+06 | $C_{25}H_{42}O_7$ |
| M47 | 28.808 | [M-H]-1 | -0.54 | Cryptochlorogenic acid | Dehydration, Nitro Reduction, Reduction | 121.06441,193.04964,149.05940 | 308.12582 | 307.11855 | 0 | 5.67E+06 | $C_{16}H_{20}O_6$ |
| | | | | Chlorogenic acid | Dehydration, Nitro Reduction, Reduction | | | | | | |
| M48 | 33.086 | [M+H]+1 | -1.05 | Caffeic acid | Desaturation, Nitro Reduction, Stearyl Conjugation | 119.08560,91.05420, 117.06995 | 414.31296 | 415.32024 | 0 | 7.71E+06 | $C_{27}H_{42}O_3$ |
| | | | | cis-4-coumaric acid | Dehydration, Reduction, Stearyl Conjugation | | | | | | |
| M49 | 37.562 | [M-H]-1 | -0.80 | Protocatechuic acid; Protocatechuic aldehyde | Reduction, Stearyl Conjugation | 375.28998,393.30219 | 422.30289 | 421.29561 | 0 | 7.92E+06 | $C_{25}H_{42}O_5$ |

their metabolism under in *vivo* conditions. Studies have shown that during metabolism, phenolic acids undergo a coupling of reactions such as methylation, sulphation and glucuronidation under the control of specific enzymes after absorption in the gastrointestinal tract [52], similar to the experimental results.

To elucidate the identification process of metabolites, we'll take M21 as an example. It only appears in the serum of the OS group, and it's noteworthy that the compounds identified in the serum have the potential to transform into one another. For instance, Rosmarinic acid is a combination of one Danshensu molecule and one caffeic acid molecule [53]. Lithospermic acid can be considered a polymerized product of one Rosmarinic acid molecule and one Danshensu molecule [54]. Salvianolic acid B is a polymer of three Danshensu molecules and one caffeic acid molecule [55], while Salvianolic acid C may be seen as a polymer of one rosmarinic acid molecule and one caffeic acid molecule [56]. Hence, during the decoction process, these compounds interconvert and function as metabolites. The precursor ion $[M-H]^-$ of M21 occurs at m/z342.09508, with the key product ions being m/z119.04882 $[M-H- C_7H_{11}O_8]^-$ and m/z 59.01249 $[M-H-C_{13}H_{15}O_7]^-$. The M21 precursor ions display a decrease of 17Da in comparison to Rosmarinic acid, an increase of 144Da compared to Danshensu, a decrease of 195Da relative to Lithospermic acid, a decrease of 375Da from Salvianolic acid B, and a decrease of 116Da in relation to Salvianolic acid C. As illustrated in Fig 9 (A1), the characteristic fragmentations of M21 correspond closely with these compounds. Therefore, based on its associated mass spectrometric cleavage behavior, M21 is considered to be a product of the Rosmarinic acid's dehydration. During the boiling process or subsequent blood entry, Danshensu, Lithospermic acid, Salvianolic acid B, and Salvianolic acid C have the potential to transform into rosmarinic acid through various pathways [57]. It has been stated in the literature that the main metabolic pathways involved in rutin in vivo are methylation, glucuronidation, sulphate esterification and their complex reactions [58]. M26 and M29 are metabolites produced by different metabolic reactions of rutin. The precursor ion $[M+K]^+$ of M26 occurs at m/z 393.07917, It is possible that a brassinose group ($C_{12}H_{20}O_9$) was first lost, and the reaction was followed by oxidation, methylation and continued loss of $CH_2O_2$. The precursor ion [M-H] + of M29 occurs at m/z 541.02960, The prototypical rutin first binds a glucuronic acid and oxidation reaction at m/z 477 based on the presence of a sulphuric acid and a methyl group at m/z 542 later, in accordance with the literature [58]. Additionally, after comparison with the specific molecular weight of Sinensetin, it was inferred that there was addition of glucuronic acid to the prototype. Mass spectrometry analysis revealed that m/z343.08109 $[M+H-C_7H_{11}O_6]^+$ and 315.08580 $[M+H-C_8H_{11}O_7]^+$ were in alignment with the fragment ions of Sinensetin. Hence, M38 was identified as the metabolite resulting from the second phase of the metabolic reaction of Sinensetin's prototype component (Fig 9 (B2)). Other metabolites were identified in a similar manner, and they're all present in the drug-containing serum alone.

## 3.4 Spatial distribution of kidney-entering components of OS in kidney tissue

Seven days subsequent to the oral administration of the OS decoction, kidney samples were extracted from the rats, precisely two hours post-administration. These samples were preserved in a refrigerator at -80˚C in preparation for performing frozen sections, with Superfrost™ Plus Slides utilized to prevent tissue detachment. The all samples from total rats in control group and OS group (n = 3) were examined using spatial mass spectrometry imaging. For spatial imaging, AFADESI-MSI mass spectrometry imaging was employed, and compounds which search by precise molecular mass for compounds confirmed by standards were identified via spatial distribution imaging, as depicted in Fig 10. Only rosemarinic acid, cis-

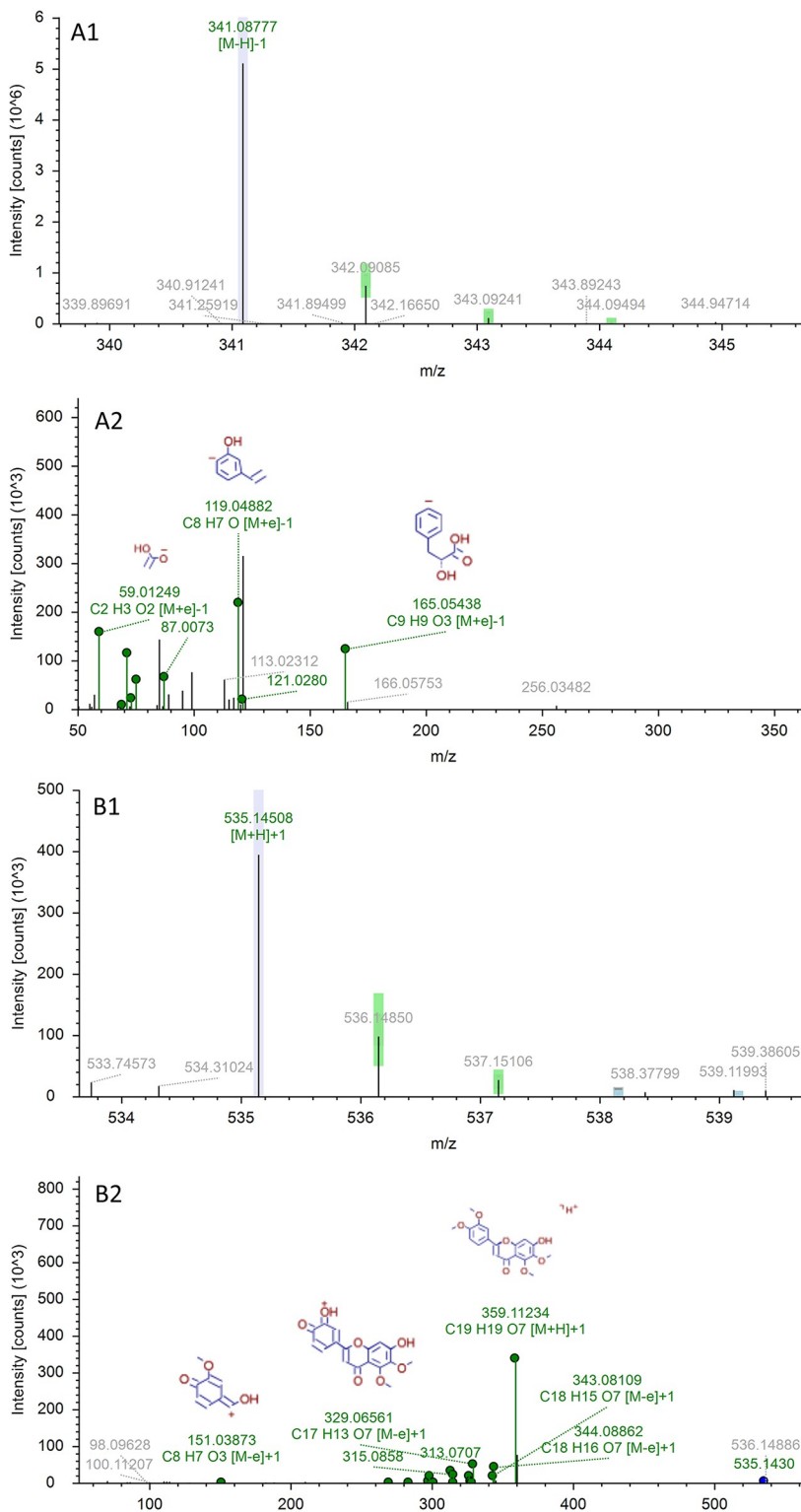

**Fig 9.** Product ions of prototype components in OS (A) and metabolites in rat plasma (B) obtained by UHPLC–Q Exactive Orbitrap–HRMS. A1: Lithospermic acid, Rosmarinic acid, Danshensu, Salvianolic Acid C, Salvianolicacid B in OS; A2: M21 in rat plasma; B1: Sinensetin in OS; B2: M38 in rat plasma.

# Rosmarinic acid

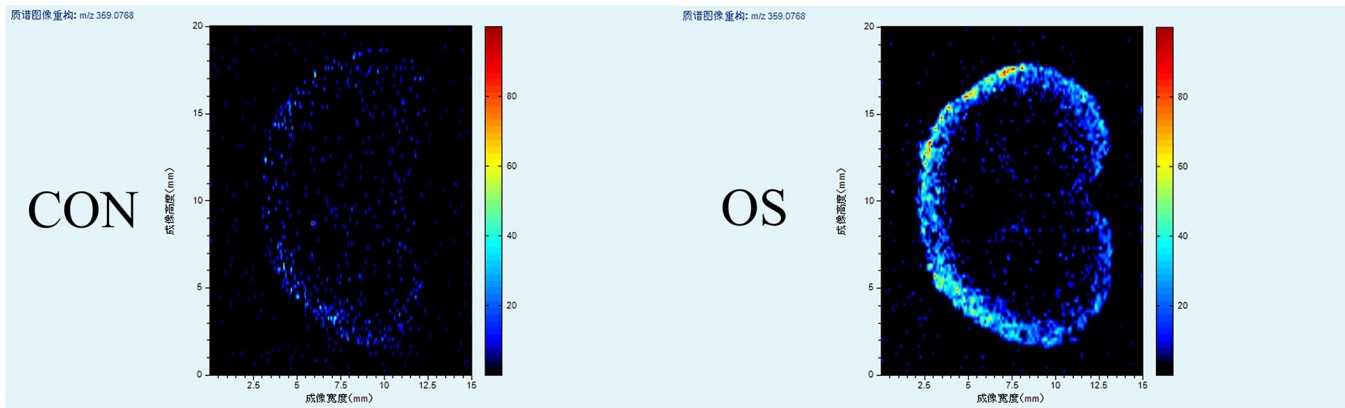

# cis-4-coumaric acid

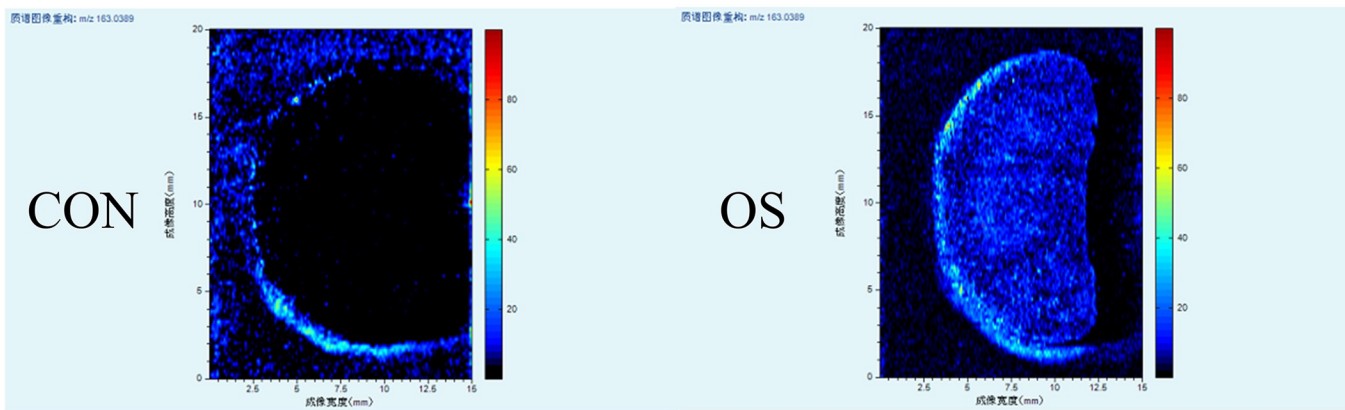

# ferulic acid

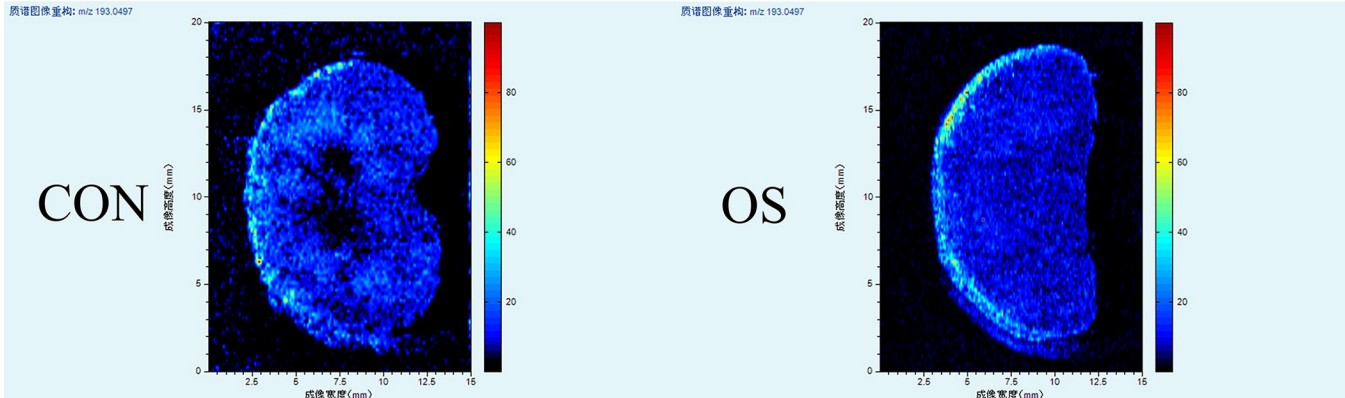

**Fig 10. Renal tissue distribution of Rosmarinic acid, cis-4-coumaric acid and ferulic acid in blank and administered OS.**

4-coumaric acid and ferulic acid demonstrated significant distributional differences in the MSI results, and no significant distribution in the control group (The remaining images are shown in S4 Fig). Predominantly, Rosmarinic acid was observed in the renal cortex following

administration, suggesting its primary action occurs in the cortex. Rosmarinic acid has been demonstrated to possess renal protective properties and has associations with oxidative damage, inflammation, and the SIRT1 and Nrf2/HO-1 signal transduction pathways [59, 60]. Studies have revealed that Rosmarinic acid can inhibit the proliferation of mesangial cells induced by cytokines [61]. Considering that glomerular mesangial cells are largely distributed between glomerular capillary loops, this evidence aligns with their spatial distribution. This can provide a foundation for subsequent studies into the mechanism of OS. Conversely, cis-4-coumaric acid was evenly distributed across the entire kidney without any distinctive tendency. While ferulic acid was reportedly present in normal kidney tissue, as it is widely found in food sources such as soybeans and potentially in rat feed, additional ferulic acid was detected in the renal calyx and renal pelvis following the administration of OS. This suggested that OS contains ferulic acid, thereby enhancing its excretion.

## 4. Conclusion

In this study, a UHPLC–Q Exactive Orbitrap–HRMS method with high sensitivity was developed to determine the underlying material basis. Active components were identified by analyzing the absorption of prototype components in rat plasma post oral OS decoction. Subsequently, a total of 92 compounds, encompassing a range from flavonoids, phenylpropanoids, other phenols, benzaldehyde derivatives, to coumarins, lignans, lignin, terpenes along with pyrrolidine alkaloids, purine alkaloids, amino acids, oligosaccharides, pyrans, cinnamic acids, amides, anthraquinones, glycosides, and stilbene glycosides, were preliminarily identified using standard materials, literature references, and databases. Certain compounds that were absent in the OS decoction led to 38 compounds being eliminated based on established standards. An analysis of the difference in compounds between OS powder and water decoction revealed the presence of more compounds post-decoction, introducing 44 new compounds, which underscores the importance of traditional Chinese medicine's decoction process. In addition, we identified 44 blood-absorbed prototype components and 50 metabolites of OS from rat serum, along with 28 prototype components within kidney tissue homogenate. Synergizing with space mass spectrometry imaging techniques allowed us to discover distribution differences of rosmarinic acid, p-coumaric acid, and ferulic acid, indicating that rosmarinic acid acts principally in the renal cortex. This finding contributes a theoretical foundation for subsequent research. This is the inaugural study to fully characterize the chemical composition of OS from the methanol extract of decoction and powder, blood, and kidney tissue in conjunction with space mass spectrometry imaging. Among them, flavonoids and phenylpropanoids accounted for the largest proportion in the OS decoction. The primary metabolic pathways consisted of hydration, dehydration, oxidation, glucuronide conjugation, nitro reduction, methylation, sulfation, and acetylation. These novel data provide a more holistic understanding of the pharmacodynamic material basis of OS. Furthermore, this study will aid in the pharmacological analysis of hyperuricemia guide network, discovery of potential drug targets, and elucidation of the mechanism and role of OS in treating hyperuricemia. It provides a theoretical basis for further exploration of the mechanism in the kidney which acted by other substances, such as rosemarinic acid, and can further explain the protective effect of OS on the kidney.

## Supporting information

**S1 Fig. TIC charts for repetition and blank.**
(TIF)

**S2 Fig. TIC charts for CON and OS group in serum.**
(TIF)

**S3 Fig. TIC charts for CON and OS group in kidney tissue.**
(TIF)

**S4 Fig. Supplementary maps for space mass spectrometry imaging(n = 3).**
(TIF)

**S1 Table. Reference standards purchase information.**
(DOCX)

**S2 Table. Compounds confirmed as absent are tabulated.**
(DOCX)

**S3 Table. Mean peak intensities of compounds identified in blood and kidney tissues in control and administered groups.**
(DOCX)

**S1 Graphical abstract.**
(TIF)

## Acknowledgments

Firstly, we would like to thank all of the participants who took part in the study, we are really grateful for your time. And I would also like to thank Proof Wei Mao and Chuang Li for their financial support of this project.

## Author Contributions

**Conceptualization:** Jianting Ouyang, Danyao Lin, Xuesheng Chen, Yimeng Li, Qin Liu, Delun Li, Haohao Quan, Xinwen Fu, Qiaoru Wu, Chuang Li, Yi Feng, Wei Mao.

**Formal analysis:** Jianting Ouyang, Danyao Lin, Xuesheng Chen, Yimeng Li, Qin Liu, Delun Li, Haohao Quan, Xinwen Fu, Qiaoru Wu.

**Funding acquisition:** Wei Mao.

**Investigation:** Jianting Ouyang.

**Methodology:** Jianting Ouyang, Chuang Li, Yi Feng, Wei Mao.

**Project administration:** Wei Mao.

**Resources:** Chuang Li, Wei Mao.

**Supervision:** Chuang Li, Yi Feng, Wei Mao.

**Validation:** Jianting Ouyang, Danyao Lin, Xuesheng Chen, Yimeng Li, Qin Liu.

**Writing – original draft:** Jianting Ouyang.

**Writing – review & editing:** Xiaowan Wang, Shouhai Wu, Wei Mao.

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
