## [Decision Letter · Decision Letter 0]

29 Feb 2024

PONE-D-24-03002Analysis of the chemical constituents and their metabolites in Orthosiphon stamineus Benth. via UHPLC-Q Exactive Orbitrap-HRMS and AFADESI-MSI techniquesPLOS ONE

Dear Dr. Mao,

Thank you for submitting your manuscript to PLOS ONE. After careful consideration, we feel that it has merit but does not fully meet PLOS ONE’s publication criteria as it currently stands. Therefore, we invite you to submit a revised version of the manuscript that addresses the points raised during the review process.

We look forward to receiving your revised manuscript.

Kind regards,

Andrea Mastinu

Academic Editor

PLOS ONE

“This work

was supported by National Natural Science Foundation of China

82074376 and 82374388, the Specific Fund of State Key Laboratory

of Dampness Syndrome of Chinese Medicine SZ2021ZZ50,SZ2021ZZ1001.

Guangdong Provincial Science and Technology Project

2022A1515012051.”

“This work was supported by National Natural Science Foundation of China (82074376 and 82374388), the Specific Fund of State Key Laboratory of Dampness Syndrome of Chinese Medicine (SZ2021ZZ50 and SZ2021ZZ1001). Guangdong Provincial Science and Technology Project (2022A1515012051).”

“This work

was supported by National Natural Science Foundation of China

82074376 and 82374388, the Specific Fund of State Key Laboratory

of Dampness Syndrome of Chinese Medicine SZ2021ZZ50,SZ2021ZZ1001.

Guangdong Provincial Science and Technology Project

2022A1515012051.”

Reviewers' comments:

Reviewer's Responses to Questions

**Comments to the Author**

1. Is the manuscript technically sound, and do the data support the conclusions?

Reviewer #1: Yes

Reviewer #2: Partly

2. Has the statistical analysis been performed appropriately and rigorously? 

Reviewer #1: No

Reviewer #2: No

3. Have the authors made all data underlying the findings in their manuscript fully available?

Reviewer #1: Yes

Reviewer #2: Yes

4. Is the manuscript presented in an intelligible fashion and written in standard English?

Reviewer #1: Yes

Reviewer #2: Yes

5. Review Comments to the Author

Reviewer #1: Dear Authors

The study deals with a meaningful and important question, so the topic interests certain readers in the Plos One. However, the following comments should be addressed because some parts should be revised and improved.

1. In your title, are there any differences between the terms chemical constituent and metabolite?

2. Did you use a QC sample in your LC-MS/MS analysis? If not, please explain it

3. In tables 1, 2, and 3 please write the molecular formula correctly like this C9H10O5

4. For your annotation of metabolite, add information about error mass to make sure your tentative identification is precise and accurate

Reviewer #2: This manuscript presents the results of research on the analysis of the chemical compounds content of Orthosiphon stamineus Benth (OS) using UHPLC-Q Exactive Orbitrap- 4 HRMS and AFADESI-MSI techniques. The research is a preliminary stage in utilizing OS as a source of medicine for kidney disease, so that the manuscript compiled is only limited to reporting a list of chemical compounds in OS and metabolites in serum from rats treated with OS extract. In this manuscript there is no in-depth analysis of the data obtained and no detailed discussion of the results of data analysis. Some notes on this manuscript are as follows:

1. The significance of this study only emphasizes the use of UHPLC-Q Exactive Orbitrap-4 HRMS for the analysis of metabolite content of OS and the use of AFADESI-MSI technique to visualize the spatial distribution of compounds in tissues.

2. The method did not describe which part of OS was used and how many biological replicates and technical replicates were conducted in this study.

3. The manuscript does not show data on the content of chemical compounds and metabolites in serum and kidney from rats that did not receive OS extract treatment (control group), so it cannot be compared between the content of chemical compounds and metabolites in the two treatments.

4. The discussion is only limited to page 25, lines 557 - 580. Even then there are confusing parts (lines 565-571). The author illustrates compound M21 at the beginning of the paragraph, but then suddenly compound M21 is explained with M18 with precursor found at m/z342.09508, which belongs to M21. In explaining the mechanism of change of chemical compounds, the authors did not use literatures as a reference.

5. Appendix B table starts from No. 93, instead of 1. There seems to be something missing.

6. Authors must be able to prove that the chemical analysis in OS was carried out from at least 3 biological replicates and at least 2 technical replicates.

7. Authors should add more detail discussion regarding the mechanism of metabolite changes in serum and kidney and compare between control and treatment groups.

6. PLOS authors have the option to publish the peer review history of their article (what does this mean?). If published, this will include your full peer review and any attached files.

Reviewer #1: No

Reviewer #2: No

---

## [Author Response · Author response to Decision Letter 0]

14 Apr 2024

Reviewer #1: Dear Authors

The study deals with a meaningful and important question, so the topic interests certain readers in the Plos One. However, the following comments should be addressed because some parts should be revised and improved.

1.In your title, are there any differences between the terms chemical constituent and metabolite?

Yes, chemical composition refers to the prototype of a compound measured in vitro such as in aqueous decoctions or powdered preparations, which exists as a prototype in the blood or tissues after administration, and metabolites, which are products resulting from metabolic reactions of the prototypical ingredient in the blood(1, 2). This metabolic operation consists of three distinct phases. Phase I metabolism encompasses functionalization reactions while Phase II involves a series of conjugation reactions. Phase III is denoted by transporter-mediated elimination of drugs and/or metabolites from the body, typically executed by the liver, gut, kidney, or lungs. The metabolites produced by prototypes in the metabolic reactions of phase I can be directly excreted or re-excreted after going through the metabolic reactions of phase II, while also excreted through the metabolic reactions of phase II directly.(3)

2.Did you use a QC sample in your LC-MS/MS analysis? If not, please explain it.

I did not use QC samples, in order to ensure the stability of the instrument and the reproducibility of the data, I prepared and tested the samples separately at different times, and observed their TIC plots, the overall profile and retention time did not find any significant changes, and combined with the comparison of the BLANK samples, the baseline is smooth, so it ensures that there are no residuals and the data can be reproduced. (S1 Fig).

3.In tables 1, 2, and 3 please write the molecular formula correctly like this C9H10O5

Thank you for your suggestion, I have changed the molecular formulae in Tables 1, 2 and 3 and in the appendix to the correct expressions

4. For your annotation of metabolite, add information about error mass to make sure your tentative identification is precise and accurate

Thank you for your suggestion, the Error (ppm) column in the Table 3 is the error value for the molecular mass of the metabolite, all within ±5.

Reviewer #2: This manuscript presents the results of research on the analysis of the chemical compounds content of Orthosiphon stamineus Benth (OS) using UHPLC-Q Exactive Orbitrap- 4 HRMS and AFADESI-MSI techniques. The research is a preliminary stage in utilizing OS as a source of medicine for kidney disease, so that the manuscript compiled is only limited to reporting a list of chemical compounds in OS and metabolites in serum from rats treated with OS extract. In this manuscript there is no in-depth analysis of the data obtained and no detailed discussion of the results of data analysis. Some notes on this manuscript are as follows:

1. The significance of this study only emphasizes the use of UHPLC-Q Exactive Orbitrap-4 HRMS for the analysis of metabolite content of OS and the use of AFADESI-MSI technique to visualize the spatial distribution of compounds in tissues.

Thanks to your suggestion, this paper also compares the compounds contained in the aqueous decoction of OS with those contained in the powder of OS and finds the necessity of decoctions of traditional Chinese medicines to co-analyse the chemical characterisation of OS not only in vivo and in vitro, but also at the spatial level to characterise the distribution of OS. The UHPLC-Q Exactive Orbitrap-HRMS technique was developed for the first time for the characterisation of kidney tea with good reproducibility and stability, and maximised the extraction of the compounds contained in OS.

2. The method did not describe which part of OS was used and how many biological replicates and technical replicates were conducted in this study.

The dry ground portion of the OS was used and three samples were prepared in parallel for each assay and a total of three replications were performed at different times to clarify the stability as well as the accuracy of the method. (S1 Fig.)

3. The manuscript does not show data on the content of chemical compounds and metabolites in serum and kidney from rats that did not receive OS extract treatment (control group), so it cannot be compared between the content of chemical compounds and metabolites in the two treatments.

The metabolic content (i.e. peak intensity or peak area) has been demonstrated in Table 3 and Appendix F, showing that the control serum and kidney tissue homogenate samples had low levels of the prototypical substance, flush with baseline and using the extracted ion-pairing approach as a complete peak that could appear, but was present in the drug-containing serum and kidney tissue. In contrast, the peak areas of the compound metabolites were all zero in the control group, indicating their specificity.

4. The discussion is only limited to page 25, lines 557 - 580. Even then there are confusing parts (lines 565-571). The author illustrates compound M21 at the beginning of the paragraph, but then suddenly compound M21 is explained with M18 with precursor found at m/z342.09508, which belongs to M21. In explaining the mechanism of change of chemical compounds, the authors did not use literatures as a reference.

Sorry for the confusion due to my mistake, the paragraph has always been about M21, as the numbering was changed during the writing process, here is my mistake. References are cited by accession.

5. Appendix B table starts from No. 93, instead of 1. There seems to be something missing.

There is no loss of data, I am sorry for the misunderstanding because of my numbering, the first 92 numbers are compounds present in the OS, because of the display and labelling of the standard's graphs, i.e. Figures 3 and 4, if the numbering starts from 1 it is easy to confuse, so the compounds present in the OS are numbered together with compounds that are not present.

6. Authors must be able to prove that the chemical analysis in OS was carried out from at least 3 biological replicates and at least 2 technical replicates.

Thank you for the reminder, the duplicate experimental plots have been added to the attachment. (S1-4 Fig)

7. Authors should add more detail discussion regarding the mechanism of metabolite changes in serum and kidney and compare between control and treatment groups.

Thank you for your suggestion, which has been added to the discussion section. Complemented the way of rutin produces metabolites in blood. It has been stated in the literature that the main metabolic pathways involved in rutin in vivo are methylation, glucuronidation, sulphate esterification and their complex reactions(4). M26 and M29 are metabolites produced by different metabolic reactions of rutin. The precursor ion [M+K]+ of M26 occurs at m/z 393.07917，It is possible that a brassinose group (C12H20O9) was first lost, and the reaction was followed by oxidation, methylation and continued loss of CH2O2. The precursor ion [M-H]+ of M29 occurs at m/z 541.02960，The prototypical rutin first binds a glucuronic acid and oxidation reaction at m/z 477 based on the presence of a sulphuric acid and a methyl group at m/z 542 later, in accordance with the literature(4).

 

References:

1. Liu YN, Hu MT, Qian J, Wang Y, Wang SF. Characterization of the chemical constituents of Jie-Geng-Tang and the metabolites in the serums and lungs of mice after oral administration by LC-Q-TOF-MS. Chin J Nat Med. 2021;19(4):284-94. http://doi.org/10.1016/S1875-5364(21)60028-6

2. Xu T, Li S, Sun Y, Pi Z, Liu S, Song F, et al. Systematically characterize the absorbed effective substances of Wutou Decoction and their metabolic pathways in rat plasma using UHPLC-Q-TOF-MS combined with a target network pharmacological analysis. J Pharm Biomed Anal. 2017;141:95-107. http://doi.org/10.1016/j.jpba.2017.04.012

3. Almazroo OA, Miah MK, Venkataramanan R. Drug Metabolism in the Liver. Clin Liver Dis. 2017;21(1):1-20. http://doi.org/10.1016/j.cld.2016.08.001

4. Goyal J, Verma PK. An Overview of Biosynthetic Pathway and Therapeutic Potential of Rutin. Mini Rev Med Chem. 2023;23(14):1451-60. http://doi.org/10.2174/1389557523666230125104101

---

## [Decision Letter · Decision Letter 1]

17 May 2024

Analysis of the chemical constituents and their metabolites in Orthosiphon stamineus Benth. via UHPLC-Q Exactive Orbitrap-HRMS and AFADESI-MSI techniques

PONE-D-24-03002R1

Dear Dr. Mao,

We’re pleased to inform you that your manuscript has been judged scientifically suitable for publication and will be formally accepted for publication once it meets all outstanding technical requirements.

Kind regards,

Andrea Mastinu

Academic Editor

PLOS ONE

Additional Editor Comments (optional):

Reviewers' comments:

Reviewer's Responses to Questions

**Comments to the Author**

1. If the authors have adequately addressed your comments raised in a previous round of review and you feel that this manuscript is now acceptable for publication, you may indicate that here to bypass the “Comments to the Author” section, enter your conflict of interest statement in the “Confidential to Editor” section, and submit your "Accept" recommendation.

Reviewer #2: All comments have been addressed

2. Is the manuscript technically sound, and do the data support the conclusions?

Reviewer #2: Yes

3. Has the statistical analysis been performed appropriately and rigorously? 

Reviewer #2: Yes

4. Have the authors made all data underlying the findings in their manuscript fully available?

Reviewer #2: Yes

5. Is the manuscript presented in an intelligible fashion and written in standard English?

Reviewer #2: Yes

6. Review Comments to the Author

Reviewer #2: (No Response)

7. PLOS authors have the option to publish the peer review history of their article (what does this mean?). If published, this will include your full peer review and any attached files.

Reviewer #2: No

---

## [Editor Report · Acceptance letter]

13 Jun 2024

PONE-D-24-03002R1 

PLOS ONE

Dear Dr. Mao, 

I'm pleased to inform you that your manuscript has been deemed suitable for publication in PLOS ONE. Congratulations! Your manuscript is now being handed over to our production team.

Kind regards, 

on behalf of

Dr. Andrea Mastinu 

Academic Editor

PLOS ONE